# Surface-reaction induced structural oscillations in the subsurface

Xianhu Sun[1], Wenhui Zhu[1], Dongxiang Wu[1], Chaoran Li[1], Jianyu Wang[1], Yaguang Zhu[1], Xiaobo Chen[1], Jorge Anibal Boscoboinik [2], Renu Sharma [3] & Guangwen Zhou [1]*

Surface and subsurface are commonly considered as separate entities because of the difference in the bonding environment and are often investigated separately due to the experimental challenges in differentiating the surface and subsurface effects. Using in-situ atomic-scale transmission electron microscopy to resolve the surface and subsurface at the same time, we show that the hydrogen–CuO surface reaction results in structural oscillations in deeper atomic layers via the cycles of ordering and disordering of oxygen vacancies in the subsurface. Together with atomistic calculations, we show that the structural oscillations in the subsurface are induced by the hydrogen oxidation-induced cyclic loss of oxygen from the oxide surface. These results demonstrate the propagation of the surface reaction dynamics into the deeper layers in inducing nonstoichiometry in the subsurface and have significant implications in modulating various chemical processes involving surface–subsurface mass transport such as heterogeneous catalysis, oxidation, corrosion and carburization.

[1] Department of Mechanical Engineering & Materials Science and Engineering Program, State University of New York, Binghamton, NY 13902, USA. [2] Center for Functional Nanomaterials, Brookhaven National Laboratory, Upton, NY 11973, USA. [3] Physical Measurement Laboratory, National Institute of Standards and Technology, Gaithersburg, MD 20899, USA. *email: gzhou@binghamton.edu

Metal oxides are of great importance in a wide range of technological applications including heterogeneous catalysis, optical display technology, solar energy devices, microelectronic devices, and oxidation prevention. The physical and chemical functionalities of the oxides are often intimately controlled by the interplay between the surface and subsurface states. For example, in heterogeneous catalysis a substantial fraction of industrial catalysis comprises oxidation reactions and oxidative dehydrogenation processes, in which metal oxides are widely used as an active catalyst[1–5]. The reactions typically follow the Mars and Van Krevelen (MvK) mechanism[6], in which the role of the oxide goes far beyond that of a simple inert spectator of the reaction because of the direct participation of its lattice oxygen in the reaction. In such cases, the oxide serves as a source of oxygen to form oxygenated compounds that desorb from the surface, thereby resulting in the generation of oxygen vacancies at the oxide surface[3]. These oxygen vacancies can further migrate into the subsurface because of the counter diffusion of oxygen from the subsurface to the surface, thus sustaining the selective catalysis. Therefore, the interplay between the removal of lattice oxygen at the oxide surface by the oxygenated product formation and the refilling of the oxygen vacancies at the surface by the outward diffusion of oxygen in the subsurface plays an important role in the reaction kinetics. The dissociation and regeneration of chemical bonds at the surface are thus linked with exchanges in chemical species between the surface and subsurface (and/or bulk) of the oxide.

Detailed insight into such surface-subsurface interactions is needed for better understanding of the fundamental features of the reactivity and for the design of efficient catalysts. That is, the control over the density and nature of oxygen vacancies both at the surface and in the subsurface as well as their exchanges could provide a means for tailoring the reactivity of oxide-based catalysts[3,7–9]. Unfortunately, deconvoluting the subsurface and surface effects has been a major challenge, and the subsurface and surface effects are therefore often investigated separately. The main challenges for probing the surface and subsurface interplay include the experimental difficulty in atomically and simultaneously resolving both the surface and subsurface regions and the long-standing challenges in overcoming the insulating nature of oxides that limits many surface-sensitive techniques based on the detection of charged particles such as electrons and ions. One way of avoiding charging effects is to prepare thin oxide films on conducting substrates, thereby allowing the use of surface science techniques such as Auger electron spectroscopy (AES) or X-ray photoelectron spectroscopy (XPS). Although these spectroscopic techniques are useful to investigate the composition in the near surface region, they are not structure sensitive and incapable of unambiguously differentiating between the surface and subsurface states because the detected signal intensity is a temporal superposition of signal originating from several atomic layers[10]. While other techniques such as scanning tunneling microscopy can provide surface structure information, they are typically limited to ultrahigh vacuum and lack the capability of resolving the atomic structure in the subsurface.

Transmission electron microscopy (TEM) is not subject to such limitations and offers the opportunity to study the oxide by providing precise information on the atomic scale for both the surface and subsurface. Particularly, TEM has dramatically evolved in recent years and now allows for temperature-resolved, pressure-resolved, and time-resolved study of the reaction dynamics by flowing a reactive gas in the sample area while simultaneously probing atomic structural evolution from the outermost surface layer to deeper atomic layers[11–19]. Such an ability to capture the dynamics of structural evolution under reaction conditions is particularly important for understanding catalytic reactions

because the state of the surface and subsurface of a catalyst is highly dynamic during its interaction with the surrounding.

In this work, we employ environmental TEM to dynamically resolve the atomic structural changes in both the surface and deeper regions of the CuO lattice in response to hydrogen induced surface reactions. The reaction between hydrogen and copper oxide is relevant to various catalytic reactions including the water-gas-shift reaction[20], methanol synthesis and oxidation[21,22], and oxidative dehydrogenation of alcohols[23]. In these reactions, the Cu oxide is an active catalyst and hydrogen is involved either as a reactant or a product present in the surrounding[20,21,23]. With the use of environmental TEM, here we show that the hydrogen–CuO surface reaction results in structural oscillations in the deeper region of the CuO lattice below the surface via cycles of ordering and disordering of oxygen vacancies in the subsurface region with a thickness of ≈3 nm from the outer surface (see schematic illustration of the experimental setup in Supplementary Fig. 1). Together with atomistic calculations, we show such structural oscillations in the subsurface are induced by the cyclic deoxygenation of the CuO surface via the reaction between surface oxygen and adsorbed hydrogen to form $H_2O$ molecules that desorb from the oxide surface with the concomitant formation of oxygen vacancies at the surface. By differentiating between the surface and subsurface states, these results have a broader applicability, in that they can be relevant to a wide range of phenomena that induce dynamical structural changes at the surface and their subsequent propagation to the deeper region of the bulk, such as oxidation, carburization, silicidation and nitridation, among others.

## Results

**In-situ TEM imaging of structural oscillations in CuO.** Our in-situ TEM experiments include the in-place CuO preparation via the oxidation of metallic Cu into CuO by exposing clean Cu to $O_2$ flow[13,14,24–27], followed by subsequently switching to $H_2$ flow. Figure 1 shows a time sequence of in-situ high-resolution TEM (HRTEM) images extracted from the Supplementary TEM Movie 1 revealing the reaction dynamics spanning from the outermost surface to the subsurface and then to the bulk during the exposure to $H_2$ flow at $pH_2 \approx 0.5$ Pa and $T \approx 300$ °C. In the beginning of the $H_2$ exposure, the CuO surface shows a step-terrace morphology with relatively uniform lattice contrast from the surface to the inner atomic layers, as shown in Fig. 1a. Diffractograms derived from the subsurface region (as marked with the dashed red box) in the HRTEM image in Fig. 1a confirm that the starting phase is monoclinic CuO with the $[\bar{1}\bar{1}\bar{1}]$ zone axis and the outer surface is oriented toward the $[\bar{1}10]$ direction (see Fig. 1e). In the first ≈19.2 s of the continuous $H_2$ flow, both the surface and subsurface regions remain relatively stable and there are no obvious structural changes, indicating an incubation period. After continued exposure to the $H_2$ flow, the initially uniform lattice contrast gradually transforms to a superlattice contrast in the subsurface region, as shown in Fig. 1b, while the bulk area still maintains the original CuO lattice. Alternate bright and faint contrast of atomic columns develops along the $(\bar{2}02)$ and $(\bar{2}\bar{2}0)$ planes from the surface to deeper layers, ≈3 nm below the surface. The superlattice feature is further confirmed by the diffractogram from the subsurface region (Fig. 1f), in which the additional spots associated with the superlattice periodicity are marked with yellow dashed circles. By comparing the diffractograms in Fig. 1e, f, it can be seen that the relative intensity of the fundamental $(\bar{1}10)$ spot has decreased drastically from that of the diffractogram of the initial CuO lattice.

Under the constant $H_2$ gas flow, the superlattice image contrast remains relatively stable for ≈17 s and then becomes significantly

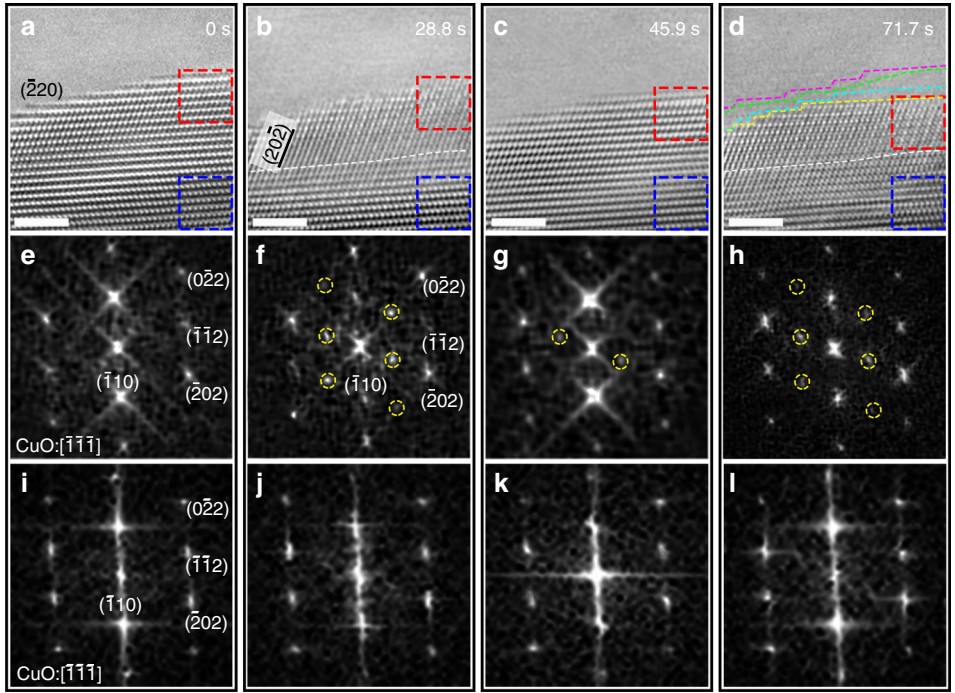

**Fig. 1 Atomic-scale visualization of structural oscillations in CuO lattice. a–d** Time-resolved HRTEM images (Supplementary Movie 1) of the CuO lattice upon exposure to a continuous flow of hydrogen gas at $T \approx 300\,°C$ and $pH_2 \approx 0.5\,Pa$. The pink, green, arctic, and yellow dashed lines in **d** are the traces of the position and configuration of the outer surface at 0 s, 28.9 s, 45.9 s, and 71.7 s, respectively, showing the oxide decay via the retraction motion of atomic steps at the outer surface. The white dashed lines in **b**, **d** mark roughly the boundary between the superlattice region and the deeper unaffected region. **e–h** Diffractograms from the subsurface region as marked with the red dashed box in the HRTEM images, showing the reoccurrence of superlattice diffraction spots indicated by yellow dashed circles. **i–l** Diffractograms from the deep region as indicated by blue dashed box in the HRTEM images, indicating that the deeper regions of the CuO lattice stays unaffected during the entire process. Scale bar, 2 nm (**a–d**).

weak. As shown in Fig. 1c, the relatively uniform lattice contrast appears again from the surface to the inner atomic layers, similar as the starting lattice feature seen in Fig. 1a. This is also confirmed by the diffractogram (Fig. 1g), which shows that most of the superlattice spots disappear and the intensity of the two remaining ones (marked by the yellow dashed circles in Fig. 1g has significantly weakened. Meanwhile, the intensities of the fundamental $(\bar{1}10)$ spots become much stronger again, similar to that in the diffractogram shown in Fig. 1e. Figure 1d corresponds to a HRTEM image captured after $\approx71.7\,s$ of the constant $H_2$ flow, which shows that the superlattice contrast comes back from the surface to the deeper layers. Similarly, the superlattice spots become visible along with the concomitant weakening of the fundamental $(\bar{1}10)$ spots in the diffractogram (Fig. 1h). The diffractograms in Fig. 1i–l from the deep region as indicated by blue dashed box in the HRTEM images confirm that the CuO lattice in the deeper regions stay unchanged during the entire process. It is worth mentioning that the vacuum annealing and electron beam effects on the observed structural oscillations are negligible (Supplementary Figs. 2–3).

The in-situ HRTEM imaging in Fig. 1a–d shows that only the subsurface region (with a thickness of $\approx3\,nm$ from the outer surface) undergoes the cyclic occurrence of the superlattice contrast whereas the CuO lattice in the deeper region remains unchanged. This therefore allows us to rule out the possibility of experimental artifacts for the observed structural oscillations in the subsurface region. As shown later, the cyclic occurrence of the superlattice contrast in the subsurface region is induced by the ordering and disordering of oxygen vacancies in the CuO lattice. By monitoring the appearance and disappearance of the superlattice spots in the diffractograms as shown in Fig. 1e–h, it can be determined that a full cycle of the ordering and disordering of

oxygen vacancies takes about 46 s. The absence of the superlattice lattice spots in diffractograms (Fig. 1i–l) from the deeper regions as marked by the dashed blue boxes confirms that the surface-subsurface interplay only affects the subsurface to a depth of $\approx3\,nm$. Detailed tracing of the temporal evolution of the surface profile is presented in Fig. 1d, showing that the surface undergoes slow decay by the retraction motion of surface steps. Cu atoms freed from the step-edge decay process migrate to inner surface regions and aggregate into a thin layer of Cu, as indicated by the presence of weak Moiré fringe contrast in the inner surface part of the sample (see more detail in Supplementary Fig. 4). However, the superlattice region maintains the relatively constant depth of $\approx3\,nm$ from the outer surface. This indicates that the surface decay is accompanied by dynamic propagation of the superlattice region toward the deeper layers, thereby maintaining a relatively constant depth of the subsurface region that is actively affected by the surface–subsurface interactions.

Such cyclic ordering and disordering of oxygen vacancies in the CuO lattice is observed from different samples. Figure 2a–c present in-situ HRTEM images (Supplementary Movie 2) showing the transformation of the initially uniform lattice contrast to a superlattice contrast and then to the uniform lattice contrast again in the subsurface region with a thickness of 3 nm from the outer surface within a time period of $\approx50\,s$. This is also confirmed by the diffractograms obtained from the regions marked by the dashed red boxes. By contrast, the CuO lattice in the deeper region stays unaffected during the entire process, as shown by the in-situ diffractograms obtained from the deeper region marked by the blue dashed boxes in Fig. 2a–c. Meanwhile, surface steps are also observed to undergo the retraction motion. Figure 2d–f show a sequence of in-situ HRTEM images (Supplementary Movie 3) from another sample region. Although the surface is relatively

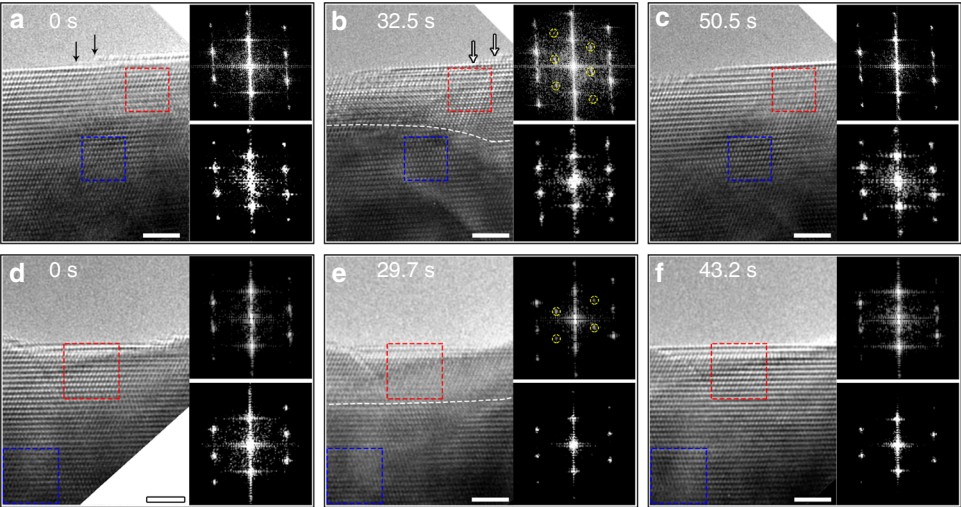

**Fig. 2 Structural oscillations in CuO lattice with different surface morphologies. a–c** Time-resolved HRTEM images (Supplementary Movie 2) of the CuO lattice upon exposure to a continuous flow of hydrogen gas at $T \approx 300\,°C$ and $pH_2 \approx 0.5\,Pa$, where the retraction motion of surface steps is monitored. Dark and white arrows in **a**, **b** mark the location of surface steps at 0 s and 32.5 s, respectively. **d–f** Time-resolved HRTEM images (Supplementary Movie 3) of the CuO lattice at $T \approx 300\,°C$ and $pH_2 \approx 0.5\,Pa$, where the majority of the surface is nearly atomically flat. Diffractograms on the upper- and lower-right side are extracted from the subsurface and deeper bulk regions as marked by the red and blue dashed boxes, respectively, in each HRTEM image. Superlattice diffraction spots are indicated by yellow dashed circles. The white dashed lines in **b**, **e** mark approximately the boundary between the superlattice region and the deeper, unaffected region. Scale bar, 2 nm.

atomically flat and does not show surface decay, the transformation from the initially uniform lattice contrast to the superlattice contrast and then again to the uniform lattice contrast is still observed in the subsurface region (3 nm thick) within a time period of ≈43 s. Similarly, the CuO lattice in the deeper region maintains unchanged during the process, as confirmed by the in-situ diffractograms.

Our in-situ TEM observations (Figs. 1–2 and Supplementary Figs. 2–4) made from different sample regions reveal some common features of the structure oscillations in the CuO lattice. These include a similar oscillation period (≈43–50 s per cycle) and a relatively constant thickness (≈3 nm) of the subsurface region that is actively affected by the surface-subsurface interplay, irrespective of the difference in the surface morphology (stepped vs. flat) and orientations (($\bar{2}20$) vs. ($\bar{3}12$)) as well as the decay motion of atomic steps. These results obtained from the different samples are mutually consistent and deliver strong evidence for the surface reaction induced ordering and disordering of oxygen vacancies in the subsurface region. These observations suggest that the actively affected zone by the surface-subsurface interplay is not tied to a specific terrace-step configuration. This is because the formation of oxygen vacancies in the CuO lattice is dominated with hydrogen adsorption by terraces rather than by surface steps. The surface restructuring during the hydrogen exposure has a negligible effect on the CuO lattice oscillations in the subsurface region, which can be attributed to the dramatic difference in hydrogen adsorption between steps and terraces. As shown in Figs. 1, 2, the surface restructuring occurs mainly through the decay of surface steps. Surface steps are known more favorable for gas adsorption than terraces, as shown by $H_2$ adsorption induced step decay at the $Cu_2O$ surface[28]. As illustrated here, hydrogen adsorption at step edges of the CuO surface induces oxygen loss and destabilizes Cu atoms within the step edge, thereby resulting in the retraction motion of atomic steps. In this process, the hydrogen adsorption at step edges does not result in oxygen vacancies due to the collapse of the entire oxide lattice along the step edge (i.e., decay of the steps). This is in contrast to hydrogen adsorption by terraces, where surface vacancies formed by the reaction between adsorbed H and lattice

O to form $H_2O$ molecules desorbing from the surface can survive due to the higher coordination at the terrace than step edges. These terrace vacancies subsequently diffuse into the subsurface region and participate in the cyclic ordering and disordering of oxygen vacancies in the CuO lattice. The absence of lattice oscillations in the deeper, bulk region can be attributed to the diluted concentration of oxygen vacancies.

Our in situ TEM experiments also indicate that the observed structure dynamics depends hydrogen pressure. No structural oscillations were observed at 300 °C and 0.01 Pa of hydrogen gas flow within a time period of more than 3 min (Supplementary Fig. 5 and Supplementary Movie 4), suggesting that the observable structural oscillations require a reasonably fast generation rate of oxygen vacancies in the CuO lattice. The slow surface reaction kinetics at the low $H_2$ pressure may result in significant dilution of the concentration of oxygen vacancies across a large depth of the sample, which does not induce observable structure changes. On the other hand, it turned out to be challenging to achieve atomic-resolution, real-time TEM imaging for HRTEM experiments conducted at 300 °C and a much higher $H_2$ pressure (5–10 Pa), where significant atomic mobility, thermal drift and scattering of electrons by gas molecules in the pressurized volume affect detrimentally the image contrast and resolution that cannot be achieved in practice. However, it is reasonable to expect that a significantly higher $H_2$ pressure can result in the over reduction of the oxide lattice, for which the CuO surface can be directly reduced to metallic Cu without observable superlattice oscillations.

**Structural interpretation of the superlattice contrast in CuO.** In-situ HRTEM images (and the diffractograms) shown above demonstrate the structural oscillations in the CuO subsurface lattice structure upon the reaction with $H_2$ at the outer surface of the oxide. We first perform HRTEM simulations of perfect CuO in order to identify the nature of the superlattice contrast. We built a $(2 \times 6 \times 5)$ supercell of perfect CuO, as shown in Fig. 3a, b is a simulated HRTEM micrograph (Supplementary Fig. 6) based on the perfect CuO structure, which matches well with the

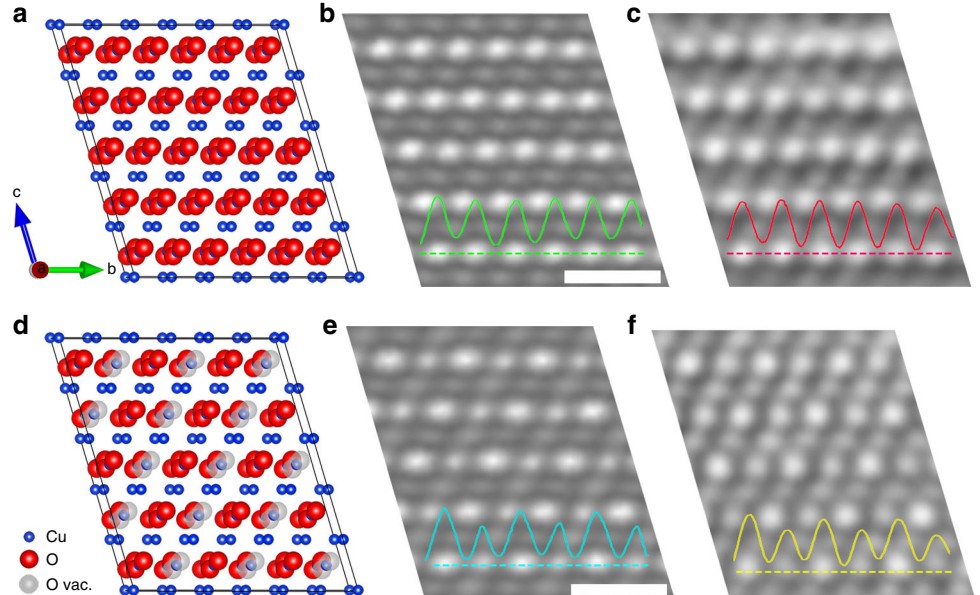

**Fig. 3 Structural interpretation of the superlattice contrast in CuO. a** Perfect CuO, viewed along the $[\bar{1}\bar{1}\bar{1}]$ direction, blue and red balls represent Cu and O atoms, respectively. **b** Simulated $[\bar{1}\bar{1}\bar{1}]$ zone-axis HRTEM micrograph of the perfect CuO. **c** Zoomed-in experimental HRTEM micrograph of the CuO lattice as shown in Fig. 1a. **d** CuO superstructure model consisting of 50% oxygen vacancies in every other Cu–O column (the positions of oxygen vacancies are represented by gray balls). **e** Simulated $[\bar{1}\bar{1}\bar{1}]$ zone-axis HRTEM micrograph based on the superlattice structure model shown in **d**. **f** Zoomed-in experimental HRTEM micrograph of the CuO superlattice shown in Fig. 1d. Insets are intensity profiles along the dashed lines in **b**, **c**, **e**, **f**, respectively. Parameters used for HRTEM image simulations are given in Methods. Scale bar, 0.5 nm (**b**, **e**).

experimental HRTEM image (Fig. 3c). It can be seen that bright and large atomic columns in the HRTEM images correspond to Cu–O columns while the weak and small columns are pure Cu columns. In perfect CuO, Cu–O columns have the same composition and therefore produce the same intensity in the HRTEM images, as seen from the intensity profiles in both the simulated and experimental images shown in Fig. 3b, c.

We then identify the origin of the superlattice contrast in the HRTEM images with atomistic modeling and HRTEM image simulations. Atomic vacancies in oxides are known to behave as atoms occupying the sites of the same lattice and typically undergo redistribution over lattice sites, giving rise to a variety of ordered structures[29–35]. As shown in Fig. 1f, h, the superlattice spots are parallel to the $(0\bar{2}2)$ and $(\bar{2}02)$ planes and the $d$-spacings between the adjacent spots in the superlattice direction are twice the $(0\bar{2}2)$ and $(\bar{2}02)$ $d$-spacings, respectively. The presence of superlattice reflections in the diffractogram provides evidence that the $H_2$ exposure results in the formation of oxygen vacancies in the CuO lattice. These oxygen vacancies self-order by condensing onto every other $(0\bar{2}2)$ and $(\bar{2}02)$ planes of the CuO lattice. Based on this information extracted from the HRTEM images in Fig. 1b, d, we built a $(2 \times 6 \times 5)$ supercell of the CuO superlattice structure model with oxygen vacancies marked by gray balls in Fig. 3d, e is a simulated HRTEM micrograph (Supplementary Fig. 7) of the CuO-superstructure (Fig. 3d) with 50% of oxygen vacancies in every other Cu–O column (i.e., a total of 25% oxygen vacancies in the entire oxide lattice), which matches well with the contrast feature of the experimental HRTEM image (Fig. 3f). This good match can be further confirmed by the matched intensity profiles of the Cu–O columns in the simulated and experimental images in Fig. 3e, f, both of which show that the image intensity for the oxygen deficient Cu–O columns is ≈30% weaker than that of the neighboring perfect Cu–O columns. The disordering of oxygen vacancies results in random distribution of oxygen vacancies in the CuO lattice and the simulated HRTEM images from randomly

distributed O vacancies show the equal image intensity of all the Cu–O columns, similar as both the simulated and experimental HRTEM images in Fig. 3b, c.

The analysis above shows that the superlattice image contrast is induced by the ordered distribution of oxygen vacancies in the CuO lattice. Density functional theory (DFT) calculations are thus used to examine the energetics associated with the oxygen vacancy formation at different sites of the CuO lattice for the ordering of oxygen vacancies. Figure 4a–d show the most favorable sequential pathway of forming four lattice oxygen vacancies at the expense of energy penalty (oxygen vacancy formation energy) of 2.5 eV, 1.4 eV, 1.4 eV, and 0.4 eV, respectively. As seen in Fig. 4d, the monoclinic structure is still maintained in the minimum-energy structure with 50% oxygen vacancies in every other Cu–O column. Figure 4e is an extended three-dimensional view along the $[\bar{1}\bar{1}\bar{1}]$ direction of the DFT-computed structure in Fig. 4d. It shows that oxygen vacancies (as marked by gray balls) form in every other Cu–O column of the oxygen-containing $(\bar{2}20)$ planes and consistent with the CuO superstructure shown in Fig. 3d, f is a perspective view of two adjacent Cu–O columns marked by the dashed purple box in Fig. 4e, which shows that the intact Cu–O column has a zigzag Cu–O–Cu double bond chain configuration whereas the oxygen-deficient column forms a zigzag Cu–O–Cu single chain configuration due to the loss of 50% of oxygen atoms. The latter resembles closely the Cu–O–Cu chain configuration in the $Cu_2O$ lattice[36]. HRTEM image simulations based on the DFT computed structure in Fig. 4e produce the same lattice contrast as shown in Fig. 3e, further substantiating that the CuO-superlattice structure in Fig. 4e is the experimentally observed superlattice. This is also confirmed by simulated electron diffraction patterns (Supplementary Fig. 8) using the CuO-superlattice structure in Fig. 4e, which show the same feature as the experimental ones in Fig. 1f, h. Therefore, the experimentally observed structure oscillations in the subsurface region shown in Fig. 1 is induced by the cyclic ordering and disordering of oxygen vacancies in the CuO lattice.

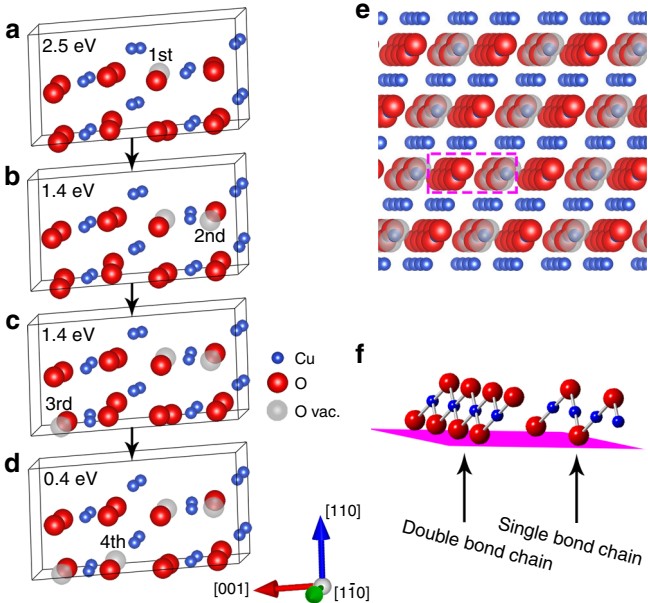

**Fig. 4 DFT modeling of oxygen vacancy formation in CuO lattice.**
**a–d** structures displaying the energetically favorable sequential pathway of oxygen loss via DFT calculations of oxygen vacancy formation energy. **e** Extended 3D structure viewed along the $[\bar{1}\bar{1}\bar{1}]$ direction of the structure in **d** with 50% oxygen vacancies in every other Cu-O column. Blue, red, and gray balls represent Cu, O, and oxygen vacancies, respectively. **f** Perspective view of two adjacent Cu-O columns marked by the dashed purple box in **e**. The intact Cu-O column has a zigzag Cu-O-Cu double bond chain configuration whereas the oxygen-deficient column has a zigzag single bond chain configuration.

**AP-XPS measurements of the H$_2$–CuO reaction.** In order to identify the source of oxygen vacancies in the subsurface region, we further employ ambient-pressure X-ray photoelectron spectroscopy (AP-XPS) to determine the chemical nature of the H$_2$-CuO surface reaction because the surface is the first place for CuO to react with H$_2$. Same as the ETEM experiments, the AP-XPS measurements also involve the two-step process by first oxidizing clean Cu to CuO in an O$_2$ gas flow[37,38], followed by switching to the flow of H$_2$ gas. Because the XPS has a much larger probed surface area ($\approx$70 μm) than ETEM, a higher pO$_2$ was used in the AP-XPS experiment to fully oxidize the Cu foil, thereby ensuring that the measured XPS signal is from the CuO surface other than from any un-oxidized Cu areas. The difference in pO$_2$ may result in differences in surface morphology and terminations of the CuO layer[39]. As shown from our in-situ TEM experiments (Figs. 1, 2 and Supplementary Figs. 2–4), the ordering and disordering of oxygen vacancies was observed in the subsurface region with a similar thickness of 3 nm for CuO surfaces with various morphologies and orientations, suggesting that the actively affected zone by the surface-subsurface interplay is not closely influenced by the surface morphology and orientation because the formation of oxygen vacancies in the CuO lattice is dominated with hydrogen adsorption by terraces rather than surface steps.

Figure 5 illustrates O 1$s$ spectra from the as-oxidized surface and during the hydrogen exposure. As shown in Fig. 5a, the O 1$s$ spectrum obtained from the as-oxidized Cu surface is relatively symmetrical with the peak positioned at 529.5 eV, which corresponds to lattice oxygen in CuO[37,38,40]. The subsequent exposure of the CuO surface to H$_2$ gas flow leads to the broadening of the O 1$s$ spectra with the presence of a large shoulder at the higher binding energy side, in which the major

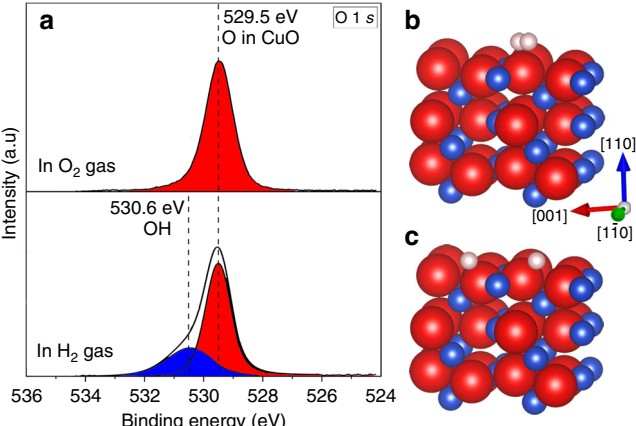

**Fig. 5 AP-XPS measurements and DFT modeling of the reaction pathway.** **a** Photoelectron spectra of O 1$s$ obtained from the CuO surface by oxidizing Cu at pO$_2 \approx 133$ Pa and $T = 350\,°C$ (upper panel) and its subsequent exposure to 0.53 Pa of the constant flow of hydrogen gas at 300 °C (lower panel). The hydrogen exposure induces the broadening of the O 1$s$ spectrum to the higher binding energy side, where the major peak corresponds to lattice oxygen in CuO (red) and the shoulder is ascribed to the presence of surface hydroxyl groups (blue). **b** A H$_2$ molecule placed above the lattice O at the CuO($\bar{2}$20) surface, **c** DFT-relaxed structure showing the spontaneous dissociation of the H$_2$ molecule into two atomic H atoms that form two OH groups by bonding to adjacent lattice O. Blue, red, and pink balls represent Cu, O, and H, respectively.

peak at 529.5 eV is still from the lattice O in CuO and the shoulder peak of 530.6 eV matches well with the binding energy of hydroxyl groups (OH)[37,41–46]. The presence of OH species provides an important clue for understanding the H$_2$–CuO reaction pathway, (i) H$_2$ → 2H, (ii) H + O → OH, (iii) OH + H → H$_2$O, where O represents lattice O at the CuO surface, OH is a stable reaction intermediate at the surface, and resultant H$_2$O molecules desorb spontaneously from the surface at the elevated temperature.

**DFT modeling of the H$_2$–CuO reaction pathway.** The reaction sequence described above is further confirmed from our atomistic modeling, as shown in Fig. 5b, c. At the beginning, one H$_2$ molecule is placed in the vacuum, $\approx$0.02 nm above the top of the lattice O at the CuO ($\bar{2}$20) surface (Fig. 5b). After the structure relaxation, the H$_2$ molecule is observed to dissociate spontaneously into two H atoms that bond with the adjacent O to form two OH groups (Fig. 5c). The spontaneous dissociation of molecular hydrogen to form OH species on the Cu oxide surface is also consistent with previous studies[28,47]. Our DFT modeling also shows that Cu sites are unfavorable for adsorption of either atomic or molecular hydrogen.

We then continue to employ DFT to examine accumulated hydrogen adsorption by the CuO surface leading to the H$_2$O formation. Figure 6 shows the minimum-energy structures with the sequential adsorption of H atoms at the energetically favorable sites of the CuO ($\bar{2}$20) surface. It can be observed that the continued adsorption of H atoms gives rise to the formation of OH groups that remain stable at the surface until reaching 0.625 monolayer (ML) of the surface coverage, at which the repulsive forces between adjacent hydroxyls make the top site of hydroxyls more favorable than the remaining sites of lattice O to adsorb further H atoms. Our calculations show that the hydrogen adsorption on top of the hydroxyls produces H$_2$O molecules that desorb from the surface without requiring any barriers. The H$_2$O desorption results in the loss of lattice oxygen with the

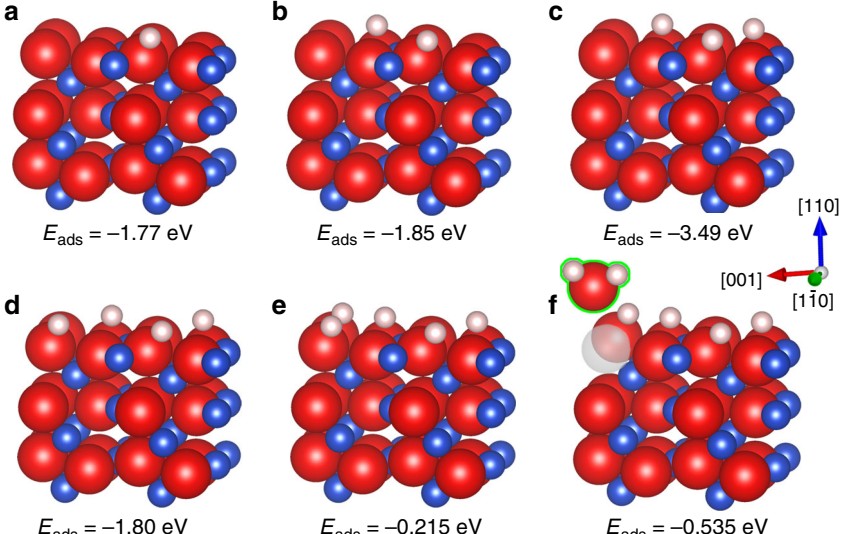

**Fig. 6 DFT modeling of H₂O formation on CuO. a–e** Minimum-energy structures of the CuO ($\bar{2}20$) surface with the sequential adsorption of H atoms to form OH groups that remain stable at the surface. **f** Adsorption of the sixth H atom results in the formation of an H₂O molecule that spontaneously desorbs from the surface, resulting in an oxygen vacancy at the surface. Blue, red, pink, and gray balls represent Cu, O, H, and O vacancy, respectively.

concomitant formation of oxygen vacancies at the oxide surface. Therefore, there is a dynamic interplay between the OH and H₂O formation, which results in the cyclic formation of oxygen vacancies at the oxide surface. The surface is dominated by the OH formation when the OH coverage is less than 0.625 ML. At the higher OH coverage, the H₂O formation becomes more favorable. Meanwhile, the continued H₂O desorption also decreases the surface coverage of OH groups, such that the OH formation becomes more favorable than the H₂O formation. This process of adsorbing H atoms to favorably increase the OH groups to the critical coverage and removing OH groups by the H₂O formation at the unfavorably higher OH coverage repeats itself and leads to the cyclic loss of lattice oxygen from the oxide surface between the low and high OH coverages. Although the formation of OH species is evidenced from the O 1s XPS spectra as shown in Fig. 5a, it is worth mentioning that the cyclic formation and removal of OH species cannot be detected readily because the measured XPS intensity is a result of the temporal and spatial summation of the overall signals from the total surface area. This is different from the in situ TEM observations as shown in Figs. 1 and 2, where the local structural dynamics can be temporally and spatially resolved.

The surface oxygen vacancies produced from the H₂O formation at the oxide surface may migrate into the subsurface because of the counter diffusion of oxygen from the subsurface to the surface, thus sustaining the surface reaction. Using DFT and nudged elastic band (NEB) computations, we further investigate the barriers for oxygen atom-vacancy exchanges between the outermost surface layer and the second surface layer for initiating the migration of vacancies from the surface to subsurface. Figure 7a shows schematically the favorable pathway for the exchange of a surface vacancy with the nearest O atom in the second layer, which requires a small diffusion barrier of 0.59 eV. This small barrier suggests that oxygen vacancies formed at the surface upon the H₂O desorption can readily migrate into the subsurface region.

Similarly, the barriers for atom-vacancy exchanges in the bulk are evaluated to assess the kinetic feasibility for the ordering and disordering transformations of oxygen vacancies. Two nonequivalent types of atom-vacancy diffusion pathways can be identified, one in the same layer (intralayer exchanges) and the

other one between adjacent layers (interlayer exchanges), as shown in Fig. 7b. It has been suggested that diffusion barrier for oxygen diffusion in CuO is inversely proportional to the concentration of oxygen vacancy, that is, the greater the concentration, the lower the diffusion barrier[48]. Here, we calculate atom-vacancy exchange barrier for a supercell with a single vacancy that corresponds to a vacancy concentration of 0.0625. Our NEB calculations show that the energy barriers for the intralayer and interlayer diffusion are 0.58 eV and 0.92 eV, respectively. These values agree reasonably well with the experimental results of the diffusion energy barriers ranging from 1.3 eV to 0.5 eV at oxygen vacancy concentrations from 0 to 0.38[48]. Kim et al. suggested that chemically dissociated hydrogen may also enter the bulk of CuO, leading to the formation of interstitial protons[49]. In the case of the presence of interstitial H in the CuO lattice (Fig. 7c), our calculations indicate that the diffusion barriers reduce to 0.51 eV and 0.69 eV for the intralayer and interlayer atom-vacancy exchanges, respectively. Therefore, penetration of H into the CuO lattice leads to a lowering of the diffusion barriers, thereby enhancing ion migration for the ordering and disordering of oxygen vacancies. As shown from these DFT calculations, the energy barriers for the different atomic processes leading to the oxide reduction center around ≈0.6 eV. This value corroborates well with in situ X-ray diffraction measurements on the hydrogen induced CuO reduction, from which the activation energy for CuO reduction was determined to be 0.63 eV[49].

## Discussion

The in-situ TEM experiments and DFT analysis point to a linkage between hydrogen induced cyclic loss of lattice oxygen at the oxide surface and the structural oscillations in the subsurface. As shown from the DFT calculations, the hydrogen adsorption by surface oxygen results in the OH formation. During this period, the surface–subsurface mass exchanges are suppressed because there is no oxygen vacancy formation at the surface and thus no available space for subsurface oxygen diffusion to until the critical OH coverage is reached to produce oxygen vacancies on the surface. This in turn allows time for the self-ordering of oxygen vacancies in the subsurface. After the OH coverage reaches

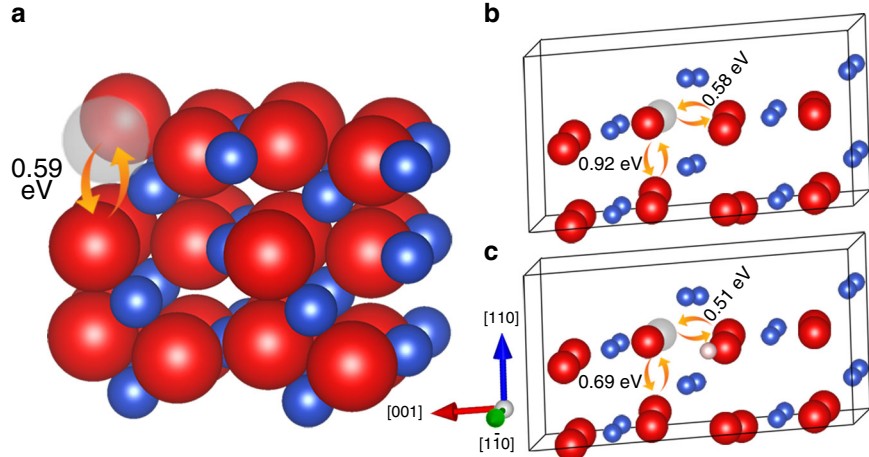

**Fig. 7 Calculations of energy barriers for atom-vacancy exchanges in CuO. a** Energy barrier for the exchange of a surface oxygen vacancy with the subsurface oxygen. **b, c** Intralayer and interlayer oxygen-vacancy diffusion barriers with and without the presence of interstitial H, respectively. Blue, red, pink, and gray balls represent Cu, O, H, and O vacancy, respectively.

0.625 ML, further hydrogen adsorption takes place via the $H_2O$ formation. Upon the desorption of $H_2O$ molecules, oxygen vacancies form at the oxide surface and subsequently migrate into the subsurface along with the outward diffusion of oxygen to refill the vacancies at the surface. The surface–subsurface mass exchanges induced by the oxygen loss from the surface trigger the disordering of the oxygen vacancies in the subsurface, thereby resulting in the disappearance of the super-lattice image contrast, as seen in Figs. 1–2. Therefore, the structural oscillations induced by the ordering and disordering of oxygen vacancies in the subsurface are closely correlated with the hydrogen adsorption induced cyclic loss of oxygen from the oxide surface. The ordering of oxygen vacancies in the subsurface results in the super-lattice formation, which is visible from the image contrast and fast Fourier transform (FFT) shown in Figs. 1, 2 and corresponds to the accumulative formation of OH groups by lattice oxygen at the surface. During this time period, no mass exchange occurs between the surface and subsurface. The disordering of oxygen vacancies occurs during the removal of OH groups from the surface via the subsequent formation and desorption of $H_2O$ molecules, and newly migrated oxygen vacancies from the surface destroy the established ordering of oxygen vacancies in the subsurface.

Metal and metal oxides are widely used as heterogeneous catalysts in industry, and their catalytic properties are intimately related to oxidation of the metallic surfaces. The presence of surface and interface oxides on metal catalysts should be taken into consideration since such species are present in the majority of real-world catalysts under reaction conditions. Their role in catalysis is still very unclear and need to be investigated on a case-by-case basis. The catalytic reactions on some transitional metals and alloys actually occur due to the surface oxidation. Recent studies on catalytic oxidation of CO and $H_2$ have suggested that the presence of a surface (or interface) oxide film such as $RuO_2$[50], CoO[51], and NiO[52] can make the catalyst catalytically more reactive than the corresponding pure metal surfaces. A microscopic understanding of such a synergistic catalytic effect requires an atomic-level understanding of the interplay between surface and subsurface states during the catalytic reactions. Particularly, oscillations in the rates of gas-surface reactions have been observed in a wide range of catalytic systems[53–58]. The identified atomistic mechanism from the model system of hydrogen oxidation over the CuO surface in the present study reveals the unique role of the surface–subsurface mass transport in modulating the fundamental steps of the surface reaction and may have broader implications for manipulating the oxide phase and non-stoichiometry to affect the reaction kinetics and mechanism.

In summary, we have identified a fundamental link between the surface reaction dynamics and structural evolution in the subsurface. Using in situ experiments and atomistic modeling, we show that the cyclic changes in hydrogen oxidation by surface oxygen and surface hydroxyls give rise to cyclic oxygen loss from the surface that, in turn, results in cycles of order-disorder transformations of oxygen vacancies in the subsurface. Non-stoichiometry occurs in a wide range of chemical compounds and is intimately connected with ordering and disordering processes of the atomic defects[7,59–61]. The cyclic ordering and disordering of atomic vacancies observed here have broader implication in modulating various gas-surface reactions. The atomistic mechanism may also find applicability in manipulating the oxide nonstoichiometry through the surface-subsurface mass transport process via the reaction dynamics at the surface.

## Methods

**In situ environmental TEM (ETEM) experiments.** In-situ experiments are divided into two parts, starting from Cu oxidation, followed by switching the gas flow from oxygen to hydrogen. Both of the processes were conducted in a dedicated ETEM equipped with an objective-lens aberration corrector and a gas manifold that enables control of the flow rate and partial pressure of various gases in the specimen area. Commercial TEM grids (99.9% purity) were used in the in situ ETEM experiments. The Cu grids were first thoroughly rinsed in deionized water followed by ultrasonication in acetone for 10 min. The Cu grids were then treated by plasma cleaning before loading into the TEM column. The Cu grids were further cleaned inside the TEM by heating to 400 °C in a $H_2$ gas flow to remove any native oxide. The cleaned Cu grids were then directly oxidized at 400 °C to form a CuO layer inside the TEM by flowing $O_2$ gas at the pressure of ≈0.5 Pa in the specimen area of the TEM column. The CuO formed from the thermal oxidation process is of both high crystallinity due to the high temperature and high purity for the elimination of any other chemical intermediaries. After the oxidation step, the specimen was then cooled down to 300 °C and the oxygen flow was stopped. Remaining oxygen in the TEM column was evacuated before hydrogen was introduced to the specimen region. The pressure of the hydrogen gas flow was maintained at ≈0.53 Pa with a flow rate of ≈0.01 SCCM (cubic centimeters per minute), and the specimen temperature was maintained at 300 °C during the hydrogen flow. This two-step process of in-place sample preparation by oxidation and subsequent observation of the reaction of the oxide with hydrogen has the advantage of minimizing potential sample contamination.

To rule out any artifacts including vacuum annealing and electron beam irradiation affecting the in situ TEM results, we performed comparison experiments, which showed that the superlattice feature is absent for the sample under the vacuum annealing whereas the superlattice contrast is observed after the sample is exposed to the $H_2$ gas flow (Supplementary Fig. 2). Similarly, the e-beam

effect was carefully minimized by adjusting the imaging condition in one area and then moving to a neighboring, fresh area for HRTEM imaging. In addition, our TEM observations by blanking and un-blanking the electron beam confirm that the electron beam effect has a negligible effect on the observed ordering and disordering of oxygen vacancies in the CuO lattice (Supplementary Fig. 3).

**Ambient-pressure X-ray photoelectron spectroscopy (AP-XPS).** AP-XPS experiments were performed at Center for Functional Nanomaterials AP-PES endstation at the CSX-2 beamline of the National Synchrotron Light Source II (NSLS-II), Brookhaven National Laboratory. The AP-XPS station is equipped with a main chamber with the base pressure lower than $6.7 \times 10^{-7}$ Pa, a hemispherical analyzer, and an Ar-ion sputtering gun. The AP-XPS system has several differential pumping stages between the reaction chamber and the hemispherical analyzer which allows keeping ultrahigh vacuum (UHV) conditions (lower than $1.3 \times 10^{-5}$ Pa) in the analyzer when the pressure in the analysis chamber is several hundreds of Pascal. Photoemitted electrons leave the high-pressure chamber through a small aperture in a conical piece into the differentially pumped transfer lenses system toward the electron energy analyzer. XPS spectra can be acquired in this system at pressures up to ≈600 Pa. The photon energy range of the beamline is from 250 to 2000 eV. Spectra of O 1$s$ were acquired in-situ at 300 °C in the presence of $H_2$ gas. Same as the ETEM experiments, Cu foils (99.9%) were used in the AP-XPS experiment with the similar treatment by rinsing in deionized water and ultra-sonication in acetone. After loading into the AP-XPS chamber, the Cu foil was further cleaned by cycles of ion sputtering and annealing to remove any native oxide. The surface cleanliness was confirmed by XPS measurements of the Cu 2$p$ and O 1$s$ peaks. The cleaned Cu was then oxidized to form a CuO layer at 400 °C by flowing $O_2$ at the pressure of ≈133 Pa in the AP-XPS chamber. After oxidation, the sample was cooled down to 300 °C and the AP-XPS chamber was evacuated. Hydrogen gas was then introduced to the chamber at a flow rate of ≈1 SCCM, where the larger flow rate for the AP-XPS system is because of its significantly larger sample compartment than ETEM. It is worth mentioning that, compared to chemical syntheses of oxide nanostructures, the CuO formed from the thermal oxidation process in our ETEM and AP-XPS experiments is of both high crystallinity (due to high temperature) and high purity (due to the elimination of any other chemical intermediaries).

**DFT calculations.** Periodic DFT calculations were performed using the Vienna Ab initio Simulation Package (VASP)[62–64]. Perdew, Burke, and Ernzerhof (PBE) generalized gradient approximation (GGA)[65] and projector augmented-wave (PAW)[66] potential were performed to describe the electron–electron exchange and core-electron potential separately. We employed DFT + $U$ in our DFT calculation since the conventional DFT functions cannot describe the strong correlation effect among the partially filled Cu 3$d$ states in CuO[67]. According to the previous study, the values of $U$ and $J$ were selected as 7 eV and 0 eV for CuO, respectively[68,69]. The plane-wave cutoff energy was set to be 400 eV for all the calculations, and spin-polarized calculations were performed since subsurface CuO has an anti-ferromagnetic ground state. For subsurface CuO, the Brillouin-zone integration was performed using ($8 \times 8 \times 8$) K-point meshes based on Monkhorst-Pack grids. A CuO supercell with 16 O and 16 Cu atoms was established for DFT calculations. CuO has the monoclinic symmetry with space group C2/c1 ($a = 0.4939$ nm, $b = 0.3674$ nm, $c = 0.5127$ nm, and $\beta = 96.152°$) after DFT + U optimization when all force components acting on the atoms were less than 0.15 eV nm$^{-1}$. Each atom has four nearest neighbors of the other kind: copper atom is located in the center of an oxygen parallelogram. Oxygen atom, in turn, is surrounded by a distorted tetrahedron of copper atoms. Lengths of the $a$-axes, $b$-axes, and $c$-axes of the supercell for simulations are fixed since the superlattice and parent phase of CuO are fully coherent, as shown in Fig. 1b, d. To study oxygen vacancy formation and hydrogen adsorption on the CuO ($\bar{2}20$) surface, a ($2 \times 2$) surface supercell expansion was used and a ($4 \times 4$) surface supercell was also used to check the results for oxygen vacancy formation. We used periodic slabs with a vacuum spacing 1.2 nm to model the CuO ($\bar{2}20$) surface, and the slab was composed of five atomic layers with the bottom two layers fixed, while the top three layers were free to relax. A ($2 \times 2$) supercell was used for modeling the CuO superlattice structure with 25% oxygen vacancies. The Brillouin zone was selected as $4 \times 4 \times 1$ and $4 \times 4 \times 4$ k-point grid for the surface and bulk models, respectively. We used $E_{ads} = \frac{1}{N_H}(E_{H/CuO}^{tot} - E_{ref} - \frac{N_O}{2}E_{H_2})$ and $E_{vac} = E_{slab/vac}^{tot} - E_{atom} - E_{slab}$ to describe the energies of hydrogen adsorption and vacancy formation, respectively. We also modeled the diffusion pathway and associated energy barriers by using the nudged elastic band (NEB) method with five intermediate images between the initial state to the final state[70]. All the atomic structures were visualized using the visualization for electronic and structure analysis (VESTA).

**HRTEM and diffractogram simulations.** Time-sequence HRTEM images in Fig. 1 were processed via Weiner filtering with 0.2 noise-to-signal ratio. The DFT-relaxed atomic structure models of CuO and CuO-superlattice were used as input files for HRTEM image and diffractogram simulations. HRTEM image simulations were performed using the multi-slice method with the parameters carefully matched to the experimental conditions (accelerating voltage: 300 keV, the spherical aberration: 0.001 mm, defocus: −1 nm, thickness: 32 nm, and Debye-Waller factors: 0.005

for both O and Cu atoms). The frozen phonon model was applied to reduce the elastic scattering and increase the background intensity.

## Disclaimer
Certain commercial equipment, instruments, or materials are identified in this paper in order to specify the experimental procedure adequately. Such identification is not intended to imply recommendation or endorsement by the National Institute of Standards and Technology, nor is it intended to imply that the materials or equipment identified are necessarily the best available for the purpose.

## Data availability
All data generated or analysed during this study are included in this published article (and its supplementary information files).

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

## Acknowledgements

This work was supported by the U.S. Department of Energy, Office of Basic Energy Sciences, Division of Materials Sciences and Engineering under Award No. DE-SC0001135. This research used resources of the Center for Functional Nanomaterials, the 23-ID-2 beamline at the National Synchrotron Light Source II, and the Scientific Data and Computing Center, a component of the Computational Science Initiative, at Brookhaven National Laboratory which is supported by the US Department of Energy, Office of Basic Energy Sciences, under Contract No. DE-SC0012704. This work also used the computational resources from the Extreme Science and Engineering Discovery Environment (XSEDE) through allocation TG-DMR110009, which is supported by National Science Foundation grant number OCI-1053575.

## Author contributions

G.Z. conceived the experiments and supervised the project. W.Z., X.S., C.L., J.W., Y.Z., X.C. and J.B. performed the experiments. D.W. conducted DFT calculations. R.S. contributed new analytic tools. X.S. and G.Z. analyzed data and wrote the paper. All the authors commented on the manuscript.

## Competing interests

The authors declare no competing interests.
