## [Peer Review File · Nature Communications]

Reviewers' comments:

Reviewer #1 (Remarks to the Author):

In the paper entitled

« Surface-reaction induced structural oscillations in the subsurface »

X. Sun et al. report the use of atomically and time resolved in situ environmental TEM to observe the atomic scale dynamics of the CuO lattice under a continuous hydrogen gas flow (at 0.53 Pa and 300 °C). Time resolved high resolution TEM images of the CuO lattice upon exposure of its surface to the flow of hydrogen reveals unexpected structural oscillations in the CuO subsurface. To identify the origin of the oscillating TEM contrasts, the authors carried out HRTEM simulations of perfect and defective CuO lattices. The combination of HRTEM images and TEM modeling and simulations reveals that the interaction between H₂ and the CuO surface results in the cyclic ordering and disordering of oxygen vacancies in the subsurface region. Combined with XPS measurements and atomistic calculations, they attribute this disorder-order structural oscillation in the CuO lattice to the cyclic loss of oxygen atoms from its surface upon hydrogen adsorption via the formation/desorption of OH groups and d H₂O molecules at the oxide (-110) surface.

The originality of this work is that it provides, for the first time, atomic scale insights into the reduction of CuO lattice by hydrogen gas. It reveals the intricate nature of the process in which the surface and subsurface work in tandem. As the surface and subsurface play an important role in the functional properties of oxides, this work should appeal to the wide scientific community working on such systems in different field of research. The analysis of the TEM and XPS data, description of the results and discussion supported by ab initio and HRTEM calculations in the submitted manuscript are clear and the manuscript is very well structured. However, to fully satisfy the criteria of excellence set by Nature Communications for publication, the following points of concern need to be addressed.

1. In figure 1, the authors state that the time serie TEM images showing the cyclic evolution of CuO lattice under constant H₂ flow are extracted from the supplementary in situ TEM video. In the latter, the CuO lattice were observed over a 5 × 14 nm² area where terraces and steps are present.

The TEM video shows that alongside the cyclic ordering and disordering of the CuO lattice, substantial surface reconstruction occurs at the CuO surface and subsurface. Do these surface defects play a role in the oxygen-vacancy order-disorder transitions? Can the authors explain why this specific region of the sample where numerous steps and terraces are present was investigated and not one with a smoother surface where the surface and subsurface region can be unambiguously defined and the structure of each more clearly established?

This question arises since it has been experimentally and theoretically shown in copper that the dissociative adsorption of molecular hydrogen is more reactive on flat Cu(111) than on stepped Cu(211) surface (Füchsel et al., *J Phys Chem Lett.* 9(1), 170–175, 2018). Similar site-dependent dissociation of H₂ molecules on the CuO (-110) cannot be ruled out in the present work. The authors should provide additional TEM observations at regions where the surface structure is less defective to show that the cyclic structural evolution of the CuO lattice shown in Figure 1 is not dependent on the presence of steps and/or terraces at the CuO surface.

Moreover, I strongly believe that, though the in situ TEM observations, presented in Figure 1, are of the highest quality, additional in situ TEM observations are needed. A single TEM observation (moreover, at a highly defective site) does not make a compelling argument for the cyclic ordering and disordering of oxygen vacancies in CuO lattice under constant H₂ flow.

2. For the TEM observation, the authors used a 300-kV electron beam. What was the corresponding electron dose?

It is known that the oxygen vacancies in nanomaterials are dynamic and can migrate under an electron beam in a TEM (for instance, see Janget al., *ACS Nano* 2017, 11, 6942–6949). Can the author comment on the strategies, if any, employed to mitigate beam effects during in situ TEM observations such that the intrinsic behavior of the CuO lattice is observed (low dose imaging, ...).

For instance, with the period of the order-disorder cycle known (from TEM observation under continuous electron illumination such as in Figure 1), an intermittent acquisition strategy can be used to confirm that the cyclic evolution of CuO lattice under H₂ prevails when the electron illumination is blanked.

3. The authors state “that alternate bright and faint contrast of atomic columns develops along the (202) and (220) planes from the surface to deeper layers, ≈ 3 nm below the surface”. On how many sublayers is the ordering and disordering of oxygen vacancies precisely observed? This number seems to vary from one region to another in Figure 1. Moreover, at a given position, does the number of layers over which the ordered structure is observed vary from one cycle to another?

4. I would expect that the ordering and disordering of oxygen vacancies observed in this work depend critically on the structure of the crystalline nature of the CuO film which is influenced by its preparation method. The authors state that they studied a copper foil but without any details about sample preparation. More details on sample preparation are needed for experiment reproducibility.

Were the copper foils used for TEM and XPS experiments similar with regards to the preparation method and pre-oxidation plasma cleaning?

5. In Gattinoni and Michaelides, *Surface Science Reports*, 424–447, 2015, it has been reported (based on DFT calculations) that CuO(111) is the most stable surface for all O₂ pressures except at very low pressures, where the Cu-terminated CuO(110) surface is stabilized.

In the XPS experiment, the Cu foil was oxidized at high O₂ pressure $p_{\text{O}_2} \approx 133$ Pa and $T=350$ °C while for the ETEM observation, the following conditions were applied to oxidize the Cu foil: $p_{\text{O}_2} \approx 0.5$ Pa and $T=400$ °C. Can the authors comment on these discrepancies? How did they ensure that the initial structure and in particular, the orientation of the surface of the copper oxide foil (before flowing H₂) in the two experiments were similar?

6. Finally, what were the H₂ flow rates during the TEM and XPS experiments?

Reviewer #2 (Remarks to the Author):

In the manuscript, authors report the surface reaction dynamics and structural evolution on CuO surface induced by adsorption of hydrogen. Using in-situ TEM, structural oscillation induced by hydrogen-CuO surface reaction was observed. It was suggested that the cycle of ordering and disordering of oxygen vacancies in the subsurface results in these structural oscillation. Atomic calculation and ambient pressure XPS were carried out to claim that the structural oscillations are induced by the hydrogen adsorption on CuO. I find the results really interesting. However, I have some doubt about the interpretation of the key observation that is oscillatory behavior of atomic structures of CuO. My comments and criticisms are shown below.

1. Hydrogen adsorption is highly surface sensitive phenomenon. However, the structural change shown in high resolution TEM images exhibits the same change of surface and bulk (up to 3 nm). For example, Figure 1a to Figure 1b, the structure of all the sample area changes. This makes me think that observation reported in the paper is not induced by the hydrogen adsorption. Maybe authors should consider other possible mechanism of structural oscillation including the instrumentational issue. Did authors observe any difference in TEM contrast between the surface and bulk? (Even low resolution TEM would reveal the difference in contrast caused by the structural evolution.)

2. The issue of mismatch between the surface and the bulk appear in the other part of the paper. HR TEM shows that surface and bulk (up to 3 nm) behave in the same way. However, the DFT modeling of hydrogen adsorption is showing the atomic structure of topmost layer.(Fig 4 and Fig 5) Authors present AP-XPS measurement of H₂-CuO reaction. Surface sensitivity of AP-XPS is determined by the electron mean free path and should be 1-2 nm. That does not exactly correspond the results shown in TEM results.

3. The surface process such as hydrogen adsorption is highly dependent upon the environment parameters including the partial pressure of hydrogen and temperature. I think if authors change the pressure or temperature during TEM experiment, the oscillatory behavior would occur in a different way. This is good evidence showing that the structural oscillations are indeed induced by the hydrogen adsorption on CuO.

Overall, the paper does not present the required level of quality and rigourousity of study to warrant publication in Nature Communications.

Response to Reviewer #1

Review report “Surface-reaction induced structural oscillations in the subsurface”

X. Sun et al. report the use of atomically and time resolved in situ environmental TEM to observe the atomic scale dynamics of the CuO lattice under a continuous hydrogen gas flow (at 0.53 Pa and 300 °C). Time resolved high resolution TEM images of the CuO lattice upon exposure of its surface to the flow of hydrogen reveals unexpected structural oscillations in the CuO subsurface. To identify the origin of the oscillating TEM contrasts, the authors carried out HRTEM simulations of perfect and defective CuO lattices. The combination of HRTEM images and TEM modeling and simulations reveals that the interaction between H₂ and the CuO surface results in the cyclic ordering and disordering of oxygen vacancies in the subsurface region. Combined with XPS measurements and atomistic calculations, they attribute this disorder-order structural oscillation in the CuO lattice to the cyclic loss of oxygen atoms from its surface upon hydrogen adsorption via the formation/desorption of OH groups and H₂O molecules at the oxide (-110) surface.

The originality of this work is that it provides, for the first time, atomic scale insights into the reduction of CuO lattice by hydrogen gas. It reveals the intricate nature of the process in which the surface and subsurface work in tandem. As the surface and subsurface play an important role in the functional properties of oxides, this work should appeal to the wide scientific community working on such systems in different field of research. The analysis of the TEM and XPS data, description of the results and discussion supported by ab initio and HRTEM calculations in the submitted manuscript are clear and the manuscript is very well structured. However, to fully satisfy the criteria of excellence set by Nature Communications for publication, the following points of concern need to be addressed.

Reply: We appreciate the encouraging words of the reviewer and his or her assessment of our results. Notably, these comments are based on a thorough and informed view of our manuscript. Please find below our answers to Reviewer’s helpful suggestions.

1. In figure 1, the authors state that the time serie TEM images showing the cyclic evolution of CuO lattice under constant H₂ flow are extracted from the supplementary in situ TEM video. In the latter, the CuO lattice were observed over a $5 \times 14 \text{ nm}^2$ area where terraces and steps are present. The TEM

video shows that alongside the cyclic ordering and disordering of the CuO lattice, substantial surface reconstruction occurs at the CuO surface and subsurface. Do these surfaces defects play a role in the oxygen-vacancy order-disorder transitions?

Reply: We greatly appreciate this insightful comment, which indeed requires elaboration. The reviewer is correct, the surface shows some restructuring by the retraction motion of surface steps. This can be evident by detailed tracing of the temporal evolution of the surface profile. However, this surface restructuring has no obvious effect on the cyclic ordering and disordering of the CuO lattice. This is confirmed from our in-situ TEM observations: 1) the thickness for the region showing the cyclic ordering and disordering is relatively constant over time (i.e., ~ 3 nm), suggesting that the surface decay is accompanied by the simultaneous inward propagation of the actively affected region toward the bulk, thereby maintaining a constant thickness (Figure 1 and in-situ TEM video 1); 2) Our additional in-situ HRTEM observations (Figure 2 and associated supplemental in-situ TEM videos 2 and 3, Supplementary Figures 2-4) further confirm that the thickness for the region displaying the ordering and disordering oscillations is similar (~ 3 nm), despite of the dramatic difference in the surface morphology and orientation; 3) the oscillation period ($\approx 43\sim 50$ s per cycle) also remains almost the same for the different sample regions and different surface morphologies.

The negligible effect of surface steps on the cyclic ordering and disordering of the CuO lattice can be attributed to the dramatic difference in hydrogen adsorption between steps and terraces. Surface steps are known to be more favorable for H_2 adsorption than terraces, as shown by H_2 adsorption induced decay of atomic steps at the Cu_2O surface (Chem. Commun., 2018, 54, 7342). As illustrated here, hydrogen adsorption at step edges of the CuO surface also induces oxygen loss and destabilizes Cu atoms within the step edge, thereby resulting in the retraction motion of atomic steps at the oxide surface. In this process, the hydrogen adsorption at step edges does not contribute to the formation of oxygen vacancies in the subsurface because of the complete collapse of the oxide lattice along steps. The decay of surface steps also results in free Cu atoms that migrate to inner surface regions and aggregate into a thin layer of Cu, as indicated by the presence of weak Moire fringe contrast in the inner part of the sample (see more detail in Supplementary Figure 4). This is in contrast to terraces, where the hydrogen adsorption induced H_2O formation results in oxygen vacancies in the surface that subsequently diffuse into the subsurface and participate in the cyclic ordering and disordering of the CuO lattice. Therefore, the observed ordering/disordering oscillations of the CuO lattice is dominated with hydrogen adsorption by terraces

that results in oxygen vacancies diffusing into the subsurface whereas the hydrogen adsorption by surface steps causes the decay of step edges without producing vacancies in the bulk.

To address this comment, we have updated Figure 1 (and Supplemental Movie 1) to include the deeper (bulk) region of the sample in the TEM images (as shown below). Although the lesser enlargement of the TEM images slightly sacrifices the crystal lattice feature in the subsurface region, it allows us to show the reaction dynamics spanning from the outermost surface to the subsurface and then to the bulk, from which we can clearly see that the subsurface region that is actively affected by the surface-subsurface interplay is ~ 3 nm deep from the outer surface, despite of the surface restructuring. In Figure 1(d), dashed lines for tracing the temporal configuration of the outer surface are added to show the step-flow oxide decay. Meanwhile, we have also included diffractograms from both the subsurface and bulk regions, which further confirm that the subsurface region undergoes cyclic ordering and disordering while the CuO lattice in the bulk remains unaffected. In addition, a new figure (given below as well) along with two new in-situ TEM movies) is added as Figure 2 to show the similar behavior of ordering and disordering of the CuO lattice in different samples with different surface morphologies. We have also added three supplementary figures (Supplementary Figures 2-4, as given below) to further confirm the superlattice feature in the subsurface region with different surface orientations and morphologies. In Supplementary Figure 3, the e-beam irradiation effect on the TEM observations is confirmed to be negligible. In Supplementary Figure 4, the presence of a thin Cu overlayer in the inner region of the sample is evidenced by the observed Moiré fringe contrast. This thin Cu overlayer is formed via the aggregation of Cu atoms released from the hydrogen adsorption induced oxide decomposition along step edges.

Figure 1: Atomic-scale visualization of the structural oscillations in the CuO lattice. (a-d): Time-resolved HRTEM images (Supplementary Movie 1) of the CuO lattice upon exposure to a continuous flow of hydrogen gas at $T \approx 300$ °C and $p_{\text{H}_2} \approx 0.5$ Pa. The pink, green, arctic, and yellow dashed lines in (d) are the traces of the position and configuration of the outer surface at 0 s, 28.9 s, 45.9 s, and 71.7 s, respectively, showing the oxide decay via the retraction motion of atomic steps at the outer surface. The white dashed lines in (b, d) mark roughly the boundary between the superlattice region and the deeper unaffected region. (e-h): Diffractograms from the subsurface region as marked with the red dashed box in the HRTEM images, showing the reoccurrence of superlattice diffraction spots indicated by yellow dashed circles. (i-l): Diffractograms from the deep region as indicated by blue dashed box in the HRTEM images, indicating that the deeper regions of the CuO lattice stays unaffected during the entire process.

Figure 2: Atomic-scale visualization of the structural oscillations in the CuO lattice with different surface morphologies. (a-c) Time-resolved HRTEM images (Supplementary Movie 2) of the CuO lattice upon exposure to a continuous flow of hydrogen gas at $T \approx 300 \text{ }^\circ\text{C}$ and $p_{\text{H}_2} \approx 0.5 \text{ Pa}$, where the retraction motion of surface steps is monitored. Dark and white arrows in (a-b) mark the location of surface steps at 0 s and 32.5 s, respectively. (d-f) Time-resolved HRTEM images (Supplementary Movie 3) of the CuO lattice at $T \approx 300 \text{ }^\circ\text{C}$ and $p_{\text{H}_2} \approx 0.5 \text{ Pa}$, where the majority of the surface is nearly atomically flat. Diffractograms on the upper- and lower-right side are extracted from the subsurface and deeper bulk regions as marked by the red and blue dashed boxes, respectively, in each HRTEM image. Superlattice diffraction spots are indicated by yellow dashed circles. The white dashed lines in (b, e) mark approximately the boundary between the superlattice region and the deeper, unaffected region.

Supplementary Figure 3: Excluding the electron beam irradiation effect on the observed structural oscillations during the H₂ exposure at T ≈ 300 °C and p_{H₂} ≈ 0.5 Pa. (a) Left: HRTEM image showing a ($\bar{3}12$) surface with the presence of the superlattice feature in the subsurface region. The dashed line marks roughly the boundary between the superlattice region and the deeper unaffected region. Middle: Diffraction pattern obtained from the subsurface marked by the red dashed box, where superlattice spots (marked by dashed yellow circles) are visible. Right: Diffraction pattern obtained from the deeper, unaffected region, as marked by the blue dashed box, in which only the fundamental spots are present. (b) Left: the e-beam was blanked for ~ 67 s and then unblanked for TEM imaging, showing that the superlattice contrast in the subsurface region has disappeared without the electron irradiation. Middle: Diffraction patterns from the subsurface region, as marked by the red dashed box, in which the superlattice spots are barely visible. Right: Diffraction pattern from the deeper region as marked by the dashed blue box, in which only the fundamental spots are visible.

Supplementary Figure 4: Aggregation of Cu atoms released from the H₂ adsorption induced decay of surface steps during H₂ exposure at T ≈ 300 °C and p_{H₂} ≈ 0.5 Pa. (a) HRTEM image showing the presence of the moiré fringe contrast as a result of the formation of a Cu overlayer on the inner surface region of the specimen via the aggregation of Cu adatoms released from the decay of atomic steps at the CuO surface. The dashed line marks roughly the boundary between the superlattice region and the deeper unaffected region. (b) Diffraction patterns from the subsurface region as marked by the dashed red box, in which superlattice spots from the ordering of oxygen vacancies are visible, as marked by the dashed yellow circles. (c) Diffraction pattern from the deeper (bulk) region, in which the superlattice spots are absent but double diffraction spots due to the overlapped lattices of Cu and CuO are visible, as marked by the small red circles.

In addition, the following sentences are incorporated into the text:

“By monitoring the appearance and disappearance of the superlattice spots in the diffraction patterns as shown in Figure 1, it can be determined that a full cycle of the ordering and disordering of oxygen vacancies takes about 46 s. The bottom panel in Figure 1 shows diffraction patterns from the deeper regions as marked by the dashed blue boxes, where the absence of the superlattice lattice spots confirm that the surface-subsurface interplay only affects the subsurface to a depth of ~ 3 nm.” (please see page 7, lines 15-20)

“Detailed tracing of the temporal evolution of the surface profile is presented in Figure 1(d), showing that the surface undergoes slow decay by the retraction motion of surface steps. However, the superlattice region maintains the relatively constant thickness of ~ 3 nm from the outer surface. This indicates that the surface decay is accompanied by the inward propagation of the superlattice region toward the deeper lattice layers, thereby dynamically maintaining a relatively constant thickness of the subsurface region that is actively affected by the surface-subsurface interactions.” (please see page 7, lines 20-28)

“Such cyclic ordering and disordering of oxygen vacancies in the CuO lattice is observed from different samples. Figures 2(a-c) present in-situ HRTEM images (Supplementary Movie 2) showing the transformation of the initially uniform lattice contrast to a superlattice contrast and then to the uniform lattice contrast again in the subsurface region with a thickness of 3 nm from the outer surface within a time period of ≈ 50 s. This is also confirmed by the diffractograms obtained from the regions marked by the dashed red boxes. By contrast, the CuO lattice in the deeper region stays unaffected during the entire process, as shown by the in-situ diffractograms obtained from the deeper region marked by the blue dashed boxes in Figures 2(a-c). Meanwhile, surface steps are also observed to undergo the retraction motion. Figures 2(d-f) show a sequence of in-situ HRTEM images (Supplementary Movie 3) from another sample region. Although the surface is relatively atomically flat and does not show surface decay, the transformation from the initially uniform lattice contrast to the superlattice contrast and then again to the uniform lattice contrast is still observed in the subsurface region (3 nm thick) within a time period of ≈ 43 s. Similarly, the CuO lattice in the deeper region maintains unchanged during the process, as confirmed by the in-situ diffractograms.” (see page 7, lines 29-33, page 8, lines 1-10)

Can the authors explain why this specific region of the sample where numerous steps and terraces are present was investigated and not one with a smoother surface where the surface and subsurface region can be unambiguously defined and the structure of each more clearly established?

Reply: As described in our response above, further analyses of the data shown in Figure 1 and additional results from other samples (Figure 2 and Supplementary Figures 2-4) show the similar feature of the cyclic ordering and disordering of oxygen vacancies in the CuO lattice, irrespective of the difference in surface morphology and orientations. This indicates that the effect from surface steps is negligible for

the observed phenomenon. This can be rationalized by the difference in hydrogen adsorption by terraces and surface steps. Hydrogen adsorption by surface steps results in the decay of the step edge, which therefore does not result in oxygen vacancies due to the collapse of the entire oxide lattice along the step edge. This is different from the hydrogen adsorption by terraces, where surface vacancies formed by the reaction between adsorbed H and lattice O can survive due to the higher coordination at the terrace than step edges. These terrace vacancies subsequently diffuse into the subsurface region and participate in the cyclic ordering disordering of the CuO lattice in the subsurface region.

We have incorporated this comment into the revision as follows:

“Our in-situ TEM observations (Figures 1-2 and Supplementary Figures 2-4) made from different sample regions reveal some common features of the structure oscillations in the CuO lattice. These include a similar oscillation period ($\approx 43\sim 50$ s per cycle) and a relatively constant thickness (≈ 3 nm) of the subsurface region that is actively affected by the surface-subsurface interplay, irrespective of the difference in the surface morphology (stepped vs. flat) and orientations ($(\bar{2}20)$ vs. $(\bar{3}12)$) as well as the decay motion of atomic steps. These results obtained from the different samples are mutually consistent and deliver strong evidence for the surface reaction induced ordering and disordering of oxygen vacancies in the subsurface region. These observations suggest that the actively affected zone by the surface-subsurface interplay is not tied to a specific terrace-step configuration. This is because the formation of oxygen vacancies in the CuO lattice is dominated with hydrogen adsorption by terraces rather than by surface steps. The surface restructuring during the hydrogen exposure has a negligible effect on the CuO lattice oscillations in the subsurface region, which can be attributed to the dramatic difference in hydrogen adsorption between steps and terraces. As shown in Figures 1 and 2, the surface restructuring occurs mainly through the decay of surface steps. Surface steps are known more favorable for gas adsorption than terraces, as shown by H_2 adsorption induced step decay at the Cu_2O surface. As illustrated here, hydrogen adsorption at step edges of the CuO surface induces oxygen loss and destabilizes Cu atoms within the step edge, thereby resulting in the retraction motion of atomic steps. In this process, the hydrogen adsorption at step edges does not result in oxygen vacancies due to the collapse of the entire oxide lattice along the step edge (i.e., decay of the steps). This is in contrast to hydrogen adsorption by terraces, where surface vacancies formed by the reaction between adsorbed H and lattice O to form H_2O molecules desorbing from the surface can survive due to the higher coordination at the terrace than step edges. These terrace vacancies subsequently diffuse into the subsurface region and participate in the cyclic ordering and disordering of oxygen vacancies in the CuO lattice. The absence of lattice oscillations in the

deeper, bulk region can be attributed to the diluted concentration of oxygen vacancies.” (please see page 9, lines 2-27)

This question arises since it has been experimentally and theoretically shown in copper that the dissociative adsorption of molecular hydrogen is more reactive on flat Cu(111) than on stepped Cu(211) surface (Füchsel et al., J Phys Chem Lett. 9(1), 170–175, 2018). Similar site-dependent dissociation of H₂ molecules on the CuO (-110) cannot be ruled out in the present work. The authors should provide additional TEM observations at regions where the surface structure is less defective to show that the cyclic structural evolution of the CuO lattice shown in Figure 1 is not dependent on the presence of steps and/or terraces at the CuO surface.

Reply: We greatly appreciate this insightful comment. Our additional in-situ TEM observations (as shown in Figure 2 and Supplementary Movies 2 and 3) confirm that surface steps at the CuO surfaces are quite active for hydrogen adsorption, which results in oxide decomposition along the step edges, as evidenced by the retraction motion of surface steps (this is also consistent with the surface decay shown in Figure 1 and Supplementary Movie 1). As described in our response above, the oxide reduction along steps results in the complete collapse of the oxide lattice and therefore does not contribute to the formation of oxygen vacancies in the subsurface region.

Moreover, I strongly believe that, though the in situ TEM observations, presented in Figure 1, are of the highest quality, additional in situ TEM observations are needed, A single TEM observation (moreover, at a highly defective site) does not make a compelling argument for the cyclic ordering and disordering of oxygen vacancies in CuO lattice under constant H₂ flow.

Reply: We are grateful to this suggestion. As described in our response above, we have followed the reviewer’s suggestion and provided additional in-situ TEM results from different samples (Figure 2, in-situ TEM videos 2 and 3, Supplementary Figures 2-4). These results obtained from different samples with different surface morphologies are mutually consistent and deliver strong evidence for the surface reaction induced ordering and disordering of oxygen vacancies in the subsurface region. These additional results are incorporated into the revision, as described above.

2. For the TEM observation, the authors used a 300-kV electron beam. What was the corresponding electron dose?

It is known that the oxygen vacancies in nanomaterials are dynamic and can migrate under an electron beam in a TEM (for instance, see Jang et al., ACS Nano 2017, 11, 6942–6949). Can the author comment on the strategies, if any, employed to mitigate beam effects during in situ TEM observations such that the intrinsic behavior of the CuO lattice is observed (low dose imaging, ...).

For instance, with the period of the order-disorder cycle known (from TEM observation under continuous electron illumination such as in Figure 1), an intermittent acquisition strategy can be used to confirm that the cyclic evolution of CuO lattice under H₂ prevails when the electron illumination is blanked.

Reply: We are appreciative of reviewer's advice on this matter. To overcome any potential e-beam effects (i.e., to rule them out as factors affecting the in-situ TEM results and to ensure that we have studied the intrinsic behavior), the e-beam effect was carefully minimized by adjusting the imaging condition in one area and then moving to a neighboring, fresh area for HRTEM imaging. In addition, we have followed the reviewer's suggestion and performed additional TEM experiments as shown below in Supplementary Figure 3. We first captured a sample area with the superlattice contrast feature in the subsurface region, as shown in Supplementary Figure 3(a). The e-beam was blanked off for about 67 s and then un-blanked for TEM imaging, which showed that the superlattice contrast has disappeared without the e-beam irradiation (Supplementary Figure 3(b)). The additional TEM observation confirms that the electron beam effect has a negligible effect on the cyclic ordering and disordering of oxygen vacancies in the CuO lattice and the phenomenon that we see is inherent. Please also note that the surface shown in Supplementary Figure 3 is terminated by the $(\bar{3}12)$ facet, which is different from the $(\bar{2}20)$ surface but still displays the superlattice feature in the subsurface region. Meanwhile, we have further examined the possible effect of vacuum annealing by performing "comparison experiments", which showed that the superlattice feature is absent for the sample under the vacuum annealing at 300 °C whereas the superlattice contrast is observed after the sample is exposed to the H₂ gas flow, as shown in Supplementary Figure 2.

In addition to include Supplementary Figures 2 and 3 in the supplementary information, the following text is also added to the experimental section to clarify this question:

“To rule out any artifacts including vacuum annealing and electron beam irradiation affecting the in-situ TEM results, we performed “comparison experiments”, which showed that the superlattice feature is absent for the sample under the vacuum annealing whereas the superlattice contrast is observed after the sample is exposed to the H₂ gas flow (Supplementary Figure 2). Similarly, the e-beam effect was carefully minimized by adjusting the imaging condition in one area and then moving to a neighboring, fresh area for HRTEM imaging. In addition, our TEM observations by blanking and un-blanking the electron beam confirm that the electron beam effect has a negligible effect on the observed ordering and disordering of oxygen vacancies in the CuO lattice (Supplementary Figure 3).” (please see page 20, lines 22-30)

Supplementary Figure 2: Atomic-scale observation of the CuO lattice under vacuum annealing and in H₂ gas flow. (a) Upper panel: HRTEM image of the CuO lattice during vacuum annealing at 300 °C. Lower

panel: Diffractograms obtained from the subsurface and deeper regions, as marked by the red and blue dashed boxes, showing the absence of superlattice spots in both regions. (b) Upper panel: HRTEM image of the CuO lattice after 30 s of H₂ gas flow at T ≈ 300 °C and p_{H₂} ≈ 0.5 Pa, where the dashed line marks roughly the boundary between the superlattice region and the deeper unaffected region. Lower panel: Diffractograms obtained from the subsurface and deeper regions, as marked by the red and blue dashed boxes, showing the subsurface transforms to the superlattice feature while the deeper region stays unaffected.

Supplementary Figure 3: Excluding the electron beam irradiation effect on the observed structural oscillations during the H₂ exposure at T ≈ 300 °C and p_{H₂} ≈ 0.5 Pa. (a) Left: HRTEM image showing a $(\bar{3}12)$ surface with the presence of the superlattice feature in the subsurface region. The dashed line marks roughly the boundary between the superlattice region and the deeper unaffected region. Middle: Diffractogram obtained from the subsurface marked by the red dashed box, where superlattice spots (marked by dashed yellow circles) are visible. Right: Diffractogram obtained from the deeper, bulk region, as marked by the blue dashed box, in which only the fundamental spots are present. (b) Left: the e-beam was blanked for ~ 67 s and then unblanked for TEM imaging, showing that the superlattice contrast in the subsurface region has disappeared without the electron irradiation. Middle: Diffractograms from the subsurface region, as marked by the red dashed box, in which the superlattice

spots are barely visible. Right: Diffractogram from the deeper region as marked by the dashed blue box, in which only the fundamental spots are visible.

3. The authors state “that alternate bright and faint contrast of atomic columns develops along the (202) and (220) planes from the surface to deeper layers, ≈ 3 nm below the surface”. On how many sublayers is the ordering and disordering of oxygen vacancies precisely observed? This number seems to vary from one region to another in Figure 1. Moreover, at a given position, does the number of layers over which the ordered structure is observed vary from one cycle to another?

Reply: As described in our response to comment #1, the thickness of the subsurface region that is actively affected by the surface-subsurface interplay is about 3 nm starting from the outer surface. This thickness is relatively constant over time, irrespective of the surface morphology, orientation and the decay motion of surface steps (Figures 1, 2 and Supplementary Figures 2-4). The absence of lattice oscillations in the deeper (bulk) region can be attributed to the diluted concentration of oxygen vacancies. In the revision, the boundary between the superlattice region and the deeper (unaffected) region is marked by a dashed line in all the TEM images (Figures 1, 2 and Supplementary Figures 2-4).

We have clarified this point in the text as follows:

“Our in-situ TEM observations (Figures 1-2 and Supplementary Figures 2-4) made from different sample regions reveal some common features of the structure oscillations in the CuO lattice. These include a similar oscillation period ($\approx 43\sim 50$ s per cycle) and a relatively constant thickness (≈ 3 nm) of the subsurface region that is actively affected by the surface-subsurface interplay, irrespective of the difference in the surface morphology (stepped vs. smooth) and orientations ($(\bar{2}20)$ vs. $(\bar{3}12)$) as well as the decay motion of atomic steps. These results obtained from the different samples are mutually consistent and deliver strong evidence for the surface reaction induced ordering and disordering of oxygen vacancies in the subsurface region. These observations suggest that the actively affected zone by the surface-subsurface interplay is not tied to a specific terrace-step configuration.” (please see page 9, lines 2-11)

“The absence of lattice oscillations in the deeper, bulk region can be attributed to the diluted concentration of oxygen vacancies.” (please see page 9, lines 26-27)

4. I would expect that the ordering and disordering of oxygen vacancies observed in this work depend critically on the structure of the crystalline nature of the CuO film which is influenced by its preparation method. The authors state that they studied a copper foil but without any details about sample preparation. More details on sample preparation are needed for experiment reproducibility.

Reply: This suggestion is well taken. For our in-situ environmental TEM experiments, commercial TEM grids (99.9% purity) were used. The Cu grids were first thoroughly rinsed in deionized water followed by ultrasonication in acetone for 10 min. The Cu grids were then treated by plasma cleaning before loading into the TEM column. The Cu grids were further cleaned inside the TEM by heating to 400 °C and flowing H₂ gas to remove any native oxide. The cleaned Cu was then directly oxidized at 400 °C to form CuO inside the TEM by flowing O₂ gas in the specimen area of the TEM column. Similarly, Cu foils (99.9%) were used in the AP-XPS experiment. The Cu foils were cleaned by rinsing in deionized water and ultrasonication in acetone. After loading into the AP-XPS chamber, the Cu foil was further cleaned by cycles of ion sputtering and annealing to remove any native oxide. The cleaned Cu was then oxidized to form a CuO layer at 400 °C by flowing O₂ at the pressure of ~ 133 Pa. It is worth mentioning that, compared to chemical syntheses of oxide nanostructures, the CuO formed from the thermal oxidation process is of both high crystallinity (due to high temperature) and high purity (due to the elimination of any other chemical intermediaries).

We have incorporated this suggestion into the experimental section as follows:

“Commercial TEM grids (99.9% purity) were used in the in-situ ETEM experiments. The Cu grids were first thoroughly rinsed in deionized water followed by ultrasonication in acetone for 10 min. The Cu grids were then treated by plasma cleaning before loading into the TEM column. The Cu grids were further cleaned inside the TEM by heating to 400 °C in a H₂ gas flow to remove any native oxide. The cleaned Cu grids were then directly oxidized at 400 °C to form a CuO layer inside the TEM by flowing O₂ gas at the pressure of 0.5 Pa in the specimen area of the TEM column. The CuO formed from the thermal oxidation process is of both high crystallinity due to the high temperature and high purity for the elimination of any other chemical intermediaries. After the oxidation step, the specimen was then cooled down to 300 °C and the oxygen flow was stopped. Remaining oxygen in the TEM column was evacuated before hydrogen was introduced to the specimen region. The pressure of the hydrogen gas flow was maintained at ≈ 0.53 Pa with a flow rate of ~0.01 SCCM (cubic centimeters per minute), and the

specimen temperature was maintained at 300 °C during the hydrogen flow.” (please see page 20, lines 6-19)

“Same as the ETEM experiments, Cu foils (99.9%) were used in the AP-XPS experiment with the similar treatment by rinsing in deionized water and ultrasonication in acetone. After loading into the AP-XPS chamber, the Cu foil was further cleaned by cycles of ion sputtering and annealing to remove any native oxide. The surface cleanliness was confirmed by XPS measurements of the Cu 2p and O 1s peaks. The cleaned Cu was then oxidized to form a CuO layer at 400 °C by flowing O₂ at the pressure of ~ 133 Pa in the AP-XPS chamber. After oxidation, the sample was cooled down to 300 °C and the AP-XPS chamber was evacuated. Hydrogen gas was then introduced to the chamber at a flow rate of ~1 SCCM, where the larger flow rate for the AP-XPS system is because of its significantly larger sample compartment than ETEM. It is worth mentioning that, compared to chemical syntheses of oxide nanostructures, the CuO formed from the thermal oxidation process in our ETEM and AP-XPS experiments is of both high crystallinity (due to high temperature) and high purity (due to the elimination of any other chemical intermediaries)” (see page 21, lines 12-25)

Were the copper foils used for TEM and XPS experiments similar with regards to the preparation method and pre-oxidation plasma cleaning?

Reply: As described in our response above, there is a slight difference in the last step of the sample preparation between the ETEM and AP-XPS experiments. In ETEM experiments, the copper foils were treated with plasma cleaning before loading into the ETEM and then further cleaned inside the TEM by annealing at 400°C in the H₂ gas flow to remove any residual native oxide. In AP-XPS experiments, the native oxide was removed by cycles of sputtering and annealing. The surface cleanliness was then confirmed by XPS measurements of Cu 2p and O 1s peaks before oxidation. We have clarified this question in the section of experimental detail, as described above.

5. In Gattinoni and Michaelides, Surface Science Reports, 424–447, 2015, it has been reported (based on DFT calculations) that CuO(111) is the most stable surface for all O₂ pressures except at very low pressures, where the Cu-terminated CuO(110) surface is stabilized. In the XPS experiment, the Cu foil was oxidized at high O₂ pressure pO₂ ≈ 133 Pa and T=350 °C while for the ETEM observation, the following conditions were applied to oxidized the Cu foil: pO₂ ≈ 0.5 Pa and T=400 °C. Can the authors

comment on these discrepancies? How did they ensure that the initial structure and in particular, the orientation of the surface of the copper oxide foil (before flowing H₂) in the two experiments were similar?

Reply: We greatly appreciate this insightful comment. Indeed, the large difference in oxygen pressure used in the oxidation may lead to different surface terminations and structures. In ETEM experiments, a lower p_{O_2} was used because of the specimen stability requirement for acquiring atomically resolved images (notably, the HRTEM imaging experiments have been performed at elevated temperature with gas flow, where significant atomic mobility, thermal drift and scattering of electrons by gas molecules in the pressurized volume can affect detrimentally the image contrast and resolution that can be achieved in practice). The CuO formation at the lower p_{O_2} was confirmed by HRTEM imaging and electron diffraction. As illustrated in Figures 1, 2 and Supplementary Figures 2-4, the ordering and disordering of oxygen vacancies was observed in the subsurface region with a similar thickness of ~ 3 nm for the CuO surfaces with various morphologies (rough vs. smooth) and orientations ($(\bar{2}20)$ vs. $(\bar{3}12)$), suggesting that the actively affected zone by the surface-subsurface interplay is not closely influenced by the surface morphology and orientation. This is because the formation of oxygen vacancies in the CuO lattice is dominated with hydrogen adsorption by terraces rather than by surface steps, as described in our response to comment 1.

In the AP-XPS experiments, a higher p_{O_2} was used to ensure that the Cu foil surface is fully covered by a CuO layer from the oxidation, therefore ensuring that the measured XPS signal is from the CuO surface other than from any unoxidized Cu areas because the XPS has a much larger probed surface area (~ 70 μm) than TEM. As described in our manuscript, the purpose of the AP-XPS experiments is to measure the surface chemistry and confirm the presence of OH groups, the cyclic ordering/disordering of oxygen vacancies and OH formation cannot be reliably resolved by XPS because of the temporal and spatial superposition of the XPS signals from different sample areas and time periods. Based on our TEM results from different sample areas with different surface morphologies and orientations, there should be no major influence on our results for the difference in the sample preparation between the ETEM and AP-XPS experiments.

In the revision, we have clarified this comment by adding the following sentences into the text:

“Because the XPS has a much larger probed surface area ($\sim 70 \mu\text{m}$) than ETEM, a higher $p\text{O}_2$ was used in the AP-XPS experiment to fully oxidize the Cu foil, thereby ensuring that the measured XPS signal is from the CuO surface other than from any un-oxidized Cu areas. The difference in $p\text{O}_2$ may result in differences in surface morphology and terminations of the CuO layer (**Gattinoni and Michaelides, Surface Science Reports 70, 424 (2015)**). As shown from our in-situ TEM experiments (Figures 1, 2 and Supplementary Figures 2-4), the ordering and disordering of oxygen vacancies was observed in the subsurface region with a similar thickness of $\sim 3 \text{ nm}$ for different CuO surfaces with various morphologies and orientations, suggesting that the actively affected zone by the surface-subsurface interplay is not closely influenced by the surface morphology and orientation. This is because the formation of oxygen vacancies in the CuO lattice is dominated by hydrogen adsorption by terraces rather than surface steps. (please see page 13, lines 15-20; page 14, lines 1-5)

6. Finally, what were the H_2 flow rates during the TEM and XPS experiments?

Reply: In the ETEM experiment, the H_2 flow rate was maintained at 0.01 SCCM. The H_2 flow rate for the AP-XPS experiment with the similar H_2 pressure of $\sim 0.5 \text{ Pa}$ as the ETEM was at 1 SCCM. The required larger flow rate for the AP-XPS system is because of its significantly larger sample compartment than TEM. We have clarified this question by including H_2 flow rates in the experimental section.

Response to Reviewer #2

Remarks to the Author:

In the manuscript, authors report the surface reaction dynamics and structural evolution on CuO surface induced by adsorption of hydrogen. Using in-situ TEM, structural oscillation induced by hydrogen-CuO surface reaction was observed. It was suggested that the cycle of ordering and disordering of oxygen vacancies in the subsurface results in these structural oscillations. Atomic calculation and ambient pressure XPS were carried out to claim that the structural oscillations are induced by the hydrogen adsorption on CuO. I find the results really interesting. However, I have some doubt about the interpretation of the key observation that is oscillatory behavior of atomic structures of CuO. My comments and criticisms are shown below.

Reply: We really appreciate the reviewer's insight, perspective, words of encouragement about this work. As can be seen below, we have developed a highly detailed set of experimental data that is thoroughly analyzed. The outcome of our revision by clearly addressing all the review criticisms is substantial and brings a fresh perspective on this work.

1. Hydrogen adsorption is highly surface sensitive phenomenon. However, the structural change shown in high resolution TEM images exhibits the same change of surface and bulk (up to 3 nm). For example, Figure 1a to Figure 1b, the structure of all the sample area changes. This makes me think that observation reported in the paper is not induced by the hydrogen adsorption. Maybe authors should consider other possible mechanism of structural oscillation including the instrumental issue. Did authors observe any difference in TEM contrast between the surface and bulk? (Even low resolution TEM would reveal the difference in contrast caused by the structural evolution.)

Reply: We greatly appreciate this insightful suggestion. The actively affected zone by the surface-subsurface interplay has a thickness of 3 ~ nm from the surface to the subsurface region, beyond which the CuO lattice is relatively unaffected because of the diluted concentration of oxygen vacancies in the deeper region. As described in our response to reviewer 1, we have updated Figure 1 (please see updated Figure 1 and Movie 1) to include the deeper (bulk) region of the sample as well as diffractograms from both the subsurface and deeper regions. Although the less enlargement of the TEM

images slightly sacrifices the crystal lattice feature in the subsurface region, it allows us to show the reaction dynamics spanning from the outer surface to the subsurface and to the bulk, from which we can clearly see that the CuO lattice in the subsurface region (with a thickness of ~3 nm starting from the outer surface) undergoes cyclic ordering and disordering whereas the CuO lattice in the deeper region stays unchanged. Therefore, it can be ruled out that the observed structural oscillation in the subsurface region is an artifact such as from instrumentation issues.

In addition, we have performed additional in-situ TEM experiments that further confirmed the hydrogen adsorption induced structural oscillation phenomenon from different samples (please see Figure 2, in-situ TEM Movies 2 and 3, Supplementary Figures 2-4). These results obtained from different samples with different surface morphologies and orientations are mutually consistent and deliver strong evidence for the surface reaction induced ordering and disordering of oxygen vacancies in the subsurface region.

In the revision, we have updated Figure 1 as described above and included additional TEM results in Figure 2, and Supplementary Figures 2-5. The following sentences are also incorporated into the text:

“The in-situ HRTEM imaging in Figures 1(a-d) shows that only the subsurface region (with a thickness of ≈ 3 nm from the outer surface) undergoes the cyclic occurrence of the superlattice contrast whereas the CuO lattice in the deeper region remains unchanged (Supplementary Figure 2). This therefore allows us to rule out the possibility of experimental artifacts for the observed structural oscillations in the subsurface region.” (please see page 7, lines 9-13)

“These results obtained from the different samples are mutually consistent and deliver strong evidence for the surface reaction induced ordering and disordering of oxygen vacancies in the subsurface region.” (please see page 9, lines 7-9)

To further help the reader understand the setup in our in-situ TEM experiments, the following schematic illustration is also added into the supplemental information to explain why the hydrogen adsorption induced superlattice contrast is observed only from the side facet along the edge-on direction. This is because the TEM image contrast is dominated with atoms along the beam (edge-on) direction. Although the planar surface is also exposed to hydrogen gas, any structure evolution at the planar surface and in

the subsurface region cannot be readily resolved from plan-view TEM imaging, for which the image contrast is dominated by the bulk information.

Supplementary Figure 1: Schematic illustration of in-situ TEM observations of hydrogen-adsorption induced structure evolution in CuO lattice. Dynamic structural evolution from the surface to subsurface and bulk can be visualized from the edge-on TEM imaging of the side facet of the specimen, where the specimen has a nominal thickness of ~ 50 nm along the beam direction. Hydrogen adsorption results in structural oscillations in the subsurface region with a depth of ~ 3 nm from the outermost surface. Noting that the planar surface is also exposed to hydrogen gas, but any structure evolution at the planar surface and in the subsurface region cannot be readily resolved from plan-view TEM imaging, where the image contrast is bulk-dominated.

2. The issue of mismatch between the surface and the bulk appear in the other part of the paper. HRTEM shows that surface and bulk (up to 3 nm) behave in the same way. However, the DFT modeling of hydrogen adsorption is showing the atomic structure of topmost layer. (Fig 4 and Fig 5) Authors present AP-XPS measurement of H₂-CuO reaction. Surface sensitivity of AP-XPS is determined

by the electron mean free path and should be 1-2 nm. That does not exactly correspond the results shown in TEM results.

Reply: As described in our response above, our in-situ HRTEM observations indicate that the subsurface region that is actively affected by the surface-subsurface interplay has a depth of about 3 nm from the outer surface, which is close to the probed depth by XPS. Please also note that the purpose of the AP-XPS experiments is to measure the surface chemistry and confirm the presence of OH groups, the cyclic ordering/disordering of oxygen vacancies and OH formation cannot be reliably resolved by XPS because of the large temporal and spatial superposition of the overall XPS signals from the total surface area. This is different from the *in-situ* TEM observations as shown in Figures 1-2, where the very local structural dynamics at the surface and in the subsurface region can be resolved because of the much higher spatial and temporal resolution by in-situ TEM imaging, as mentioned in the manuscript.

We have clarified this point in the manuscript as follows:

“it is worth mentioning that the cyclic formation and removal of OH species cannot be detected readily because the measured XPS intensity is a result of the temporal and spatial summation of the overall signals from the total surface area. This is different from the *in-situ* TEM observations as shown in Figures 1 and 2, where the local structural dynamics can be temporally and spatially resolved.” (page 16, lines 12-16)

3. The surface process such as hydrogen adsorption is highly dependent upon the environment parameters including the partial pressure of hydrogen and temperature. I think if authors change the pressure or temperature during TEM experiment, the oscillatory behavior would occur in a different way. This is good evidence showing that the structural oscillations are indeed induced by the hydrogen adsorption on CuO.

Reply: This is a great point! As described in our response to the comments from reviewer 1, our in-situ TEM results reveal that the structural oscillation is not tied to a specific surface morphology and orientation. Our additional in-situ TEM experiments indicate that the structure dynamics indeed depends on H₂ pressure. As shown in the following in-situ HRTEM images (Supplementary Figure 5 and Supplementary Movie 4), no structural oscillations were observed at 300°C and 0.01 Pa of hydrogen

pressure within a time period of more than 3 min. This suggests that the observable structural oscillations require a reasonably fast generation rate of oxygen vacancies in the CuO lattice. The slow surface reaction kinetics (due to the low H₂ pressure) may result in significant dilution of the concentration of oxygen vacancies across a large depth of the sample, which therefore does not induce any observable structure feature. On the other hand, it turned out to be quite challenging to perform atomic-resolution, real-time TEM imaging with a higher H₂ pressure. Our additional in-situ HRTEM experiments conducted at 300 °C and 5~10 Pa of the H₂ gas flow showed significant blurring of the TEM image contrast. This is because the HRTEM imaging experiments have been performed with the thin film specimens at elevated temperature and under the gas flow, where significant atomic mobility, thermal drift and scattering of electrons by gas molecules in the pressurized volume can affect detrimentally the image contrast and resolution that can be achieved in practice. However, it is reasonable to expect that a significantly higher hydrogen pressure can result in the over reduction of the oxide lattice, for which the CuO surface can be directly reduced to metallic Cu without observable superlattice oscillations.

Supplementary Figure 5: Time-sequence HRTEM images (Supplementary Movie 4) showing no structural oscillations within a time period of 190.5 s for a CuO sample at 300 °C and 0.01 Pa of H₂ gas flow.

In the revision, we have included these additional results in the supplementary information (Supplementary Figure 5). Meanwhile, the following sentences are also incorporated into the text:

“Our in-situ TEM experiments also indicate that the observed structure dynamics depends hydrogen pressure. No structural oscillations were observed at 300°C and 0.01 Pa of hydrogen gas flow within a time period of more than 3 min (Supplementary Figure 5), suggesting that the observable structural oscillations require a reasonably fast generation rate of oxygen vacancies in the CuO lattice. The slow surface reaction kinetics at the low H₂ pressure may result in significant dilution of the concentration of oxygen vacancies across a large depth of the sample, which does not induce observable structure changes. On the other hand, it turned out to be challenging to achieve atomic-resolution, real-time TEM imaging for HRTEM experiments conducted at 300°C and 5~10 Pa of the H₂ gas flow, where significant atomic mobility, thermal drift and scattering of electrons by gas molecules in the pressurized volume affect detrimentally the image contrast and resolution that can be achieved in practice. However, it is reasonable to expect that a significantly higher H₂ pressure can result in the over reduction of the oxide lattice, for which the CuO surface can be directly reduced to metallic Cu without observable superlattice oscillations. (See Page 9, lines 28-31; Page 10, lines 1-10)

Overall, the paper does not present the required level of quality and rigorosity of study to warrant publication in Nature Communications.

Reply: We believe that we have addressed all the issues raised, and the revised manuscript has been greatly improved as a result of your constructive comments and suggestions. We deeply appreciate your guidance in improving this work.

REVIEWERS' COMMENTS:

Reviewer #1 (Remarks to the Author):

In my previous report, I raised a number of concerns on the submitted manuscript and provided few suggestions for its improvement. These concerns have been thoroughly addressed and all suggestions considered by the authors during revision. They have implemented a number of changes and included additional experimental data which have significantly improved the manuscript. In my opinion, the manuscript in its present form is now suitable for publication in Nature Communications.

Reviewer #2 (Remarks to the Author):

I read the revised manuscript and authors' response carefully. Authors addressed my comments and criticism appropriately, therefore, the quality of manuscript has been improved significantly. Especially the point regarding the structural changes of the surface and bulk has been addressed in a convincing way due to additional TEM results.

Therefore, I suggest that the paper be accepted after minor revision.

I have one comment that can be addressed in the revision process. While the current manuscript demonstrates the cyclic ordering and disordering of atomic vacancies, the catalytic reactivity of CuO is not discussed. The recent in-situ microscopic and spectroscopic surface techniques shows that the thin oxide on metal surface could be catalytically reactive [For example, H. Lee et al. Nat Comm Nature Communications 9, 2235 (2018); J. Kim et al. Sci. Adv. 4, eaat3151 (2018)]. In relation to this point, the aspect of catalytic reactivity of CuO can be discussed.

REVIEWERS' COMMENTS:

Reviewer #1 (Remarks to the Author):

In my previous report, I raised a number of concerns on the submitted manuscript and provided few suggestions for its improvement. These concerns have been thoroughly addressed and all suggestions considered by the authors during revision. They have implemented a number of changes and included additional experimental data which have significantly improved the manuscript. In my opinion, the manuscript in its present form is now suitable for publication in Nature Communications.

Reviewer #2 (Remarks to the Author):

I read the revised manuscript and authors' response carefully. Authors addressed my comments and criticism appropriately, therefore, the quality of manuscript has been improved significantly. Especially the point regarding the structural changes of the surface and bulk has been addressed in a convincing way due to additional TEM results.

Therefore, I suggest that the paper be accepted after minor revision.

I have one comment that can be addressed in the revision process. While the current manuscript demonstrates the cyclic ordering and disordering of atomic vacancies, the catalytic reactivity of CuO is not discussed. The recent in-situ microscopic and spectroscopic surface techniques shows that the thin oxide on metal surface could be catalytically reactive [For example, H. Lee et al. Nat Comm Nature Communications 9, 2235 (2018); J. Kim et al. Sci. Adv. 4, eaat3151 (2018)]. In relation to this point, the aspect of catalytic reactivity of CuO can be discussed.

Reply: Per Reviewer's suggestion, we have added the following paragraph to discuss the broader implication of our results in understanding catalytic mechanisms and kinetics:

“Metal and metal oxides are widely used as heterogeneous catalysts in industry, and their catalytic properties are intimately related to oxidation of the metallic surfaces. The presence of surface and interface oxides on metal catalysts should be taken into consideration since such species are present in the majority of real-world catalysts under reaction conditions. Their role in catalysis is still very unclear and need to be investigated on a case-by-case basis. The catalytic reactions on some transitional metals and alloys actually occur due to the surface oxidation. Recent studies on catalytic oxidation of CO and H₂ have suggested that the presence of a surface (or interface) oxide film such as RuO₂³⁵, CoO³⁶, and NiO³⁷ can make the catalyst catalytically more reactive than the corresponding pure metal surfaces. A microscopic understanding of such a synergistic catalytic effect requires an atomic-level understanding of the interplay between surface and subsurface states during the catalytic reactions. Particularly, oscillations in the rates of gas-surface reactions have been observed in a wide range of catalytic systems³⁸⁻⁴³. The identified atomistic mechanism from the model system of hydrogen oxidation over the CuO surface in the present study reveals the unique role of the surface-subsurface mass transport in

modulating the fundamental steps of the surface reaction and may have broader implications for manipulating the oxide phase and nonstoichiometry to affect the reaction kinetics and mechanism.”

(please see paragraphs marked in red on Pages 18 and 19)